# Notch signaling is a driver of glandular stem cell activity and regenerative migration after damage

Davide Cinat [1,2,8 ✉], Rufina Maturi [2,3], Jeremy P Gunawan[1,2], Anne L Jellema-de Bruin[1,2], Laura Kracht[2,4], Paola Serrano Martinez[2,5], Yi Wu[1,2,6], Abel Soto-Gamez [1,2], Marc-Jan van Goethem[1], Inge R Holtman[2], Sarah Pringle [7], Lara Barazzuol [1,2 ✉] & Rob P Coppes [1,2 ✉]

## Abstract

Organoid models have significantly enhanced our understanding of adult stem cell function, however, uncovering regulatory mechanisms governing rare and often quiescent stem cells in glandular organs remains challenging. Here, we employ an integrative multi-omics approach, combining single-cell RNA sequencing, bulk ATAC and RNA sequencing, to profile the cellular populations and signaling pathways characterizing a mouse salivary gland organoid model across different temporal stages and after radiation-induced damage. Our findings identify *Sox9*- and *Itgb1/Cd44*-expressing cells as primitive adult stem/progenitor populations with a critical migratory role in tissue repair. Notch signaling is a key driver of self-renewal and migration in response to irradiation. Additionally, scRNA-seq analysis of irradiated salivary gland tissue confirms these findings in an in vivo setting. Extending these findings to murine and patient-derived salivary, mammary and thyroid gland organoids, we reveal the conserved role of Notch signaling in coordinating stem/progenitor cell-mediated regeneration across glandular tissues. These insights position Notch signaling as a central regulator of glandular stem cell-like populations and as a promising therapeutic target for enhancing glandular tissue regeneration following cancer therapies.

**Keywords** Glandular Organoids; Notch Signaling; Radiotherapy; Adult Stem Cells; Cell Migration
**Subject Categories** Methods & Resources; Signal Transduction; Stem Cells & Regenerative Medicine

## Introduction

Adult stem/progenitor cells in glandular tissues, like the salivary glands, are scarce and often residing in a quiescent state, making their identification challenging (Rocchi et al, 2021a). Moreover, their regenerative capacities in response to damage have been shown to be heavily influenced by the microenvironment, impacting self-renewal and differentiation abilities (De Morree and Rando, 2023; Cinat et al, 2021). Given their critical role in maintaining tissue homeostasis (Biteau et al, 2011) and the demand for personalized and regenerative therapies (Altshuler et al, 2023; Soto-Gamez et al, 2024), there has been growing interest in characterizing their identity and regulatory pathways (Altshuler et al, 2023; Biteau et al, 2011).

Salivary glands are exocrine organs composed of serous and mucous saliva-secreting acinar cells, myoepithelial cells, and a network of intercalated, luminal and basal duct cells (Rocchi et al, 2021a). Among these, the basal layer of the ductal compartment is believed to host putative stem/progenitor cells (Rocchi et al, 2021a; May et al, 2018). Although organoids have proven to be a valuable model for characterizing salivary gland stem/progenitor cells (Orhon et al, 2022; Peng et al, 2020), a comprehensive understanding of the cells constituting this organoid model and the signaling pathways regulating their regenerative functions is still lacking.

Radiotherapy is one of the most common therapeutic modalities for treating head and neck cancer (Mody et al, 2021). However, due to the close proximity to the radiation site, salivary glands often suffer collateral damage after treatment (Van Luijk et al, 2015; Mody et al, 2021), resulting in the loss of stem/progenitor cell populations and subsequent impairment of tissue regeneration (Van Luijk et al, 2015; van Rijn-Dekker et al, 2024). This can result in side effects, such as radiation-induced hyposalivation and related xerostomia, significantly impairing patients' quality of life (Mercadante et al, 2021). Salivary gland-derived organoids have played a salient role in elucidating the molecular mechanisms underlying the response to radiation-induced DNA damage (Peng et al, 2020;

[1]Department of Radiation Oncology, University Medical Center Groningen, University of Groningen, Groningen 9713 GZ, The Netherlands. [2]Department of Biomedical Sciences, University Medical Center Groningen, University of Groningen, Groningen 9713 GZ, The Netherlands. [3]Department of Molecular Medicine and Medical Biotechnology, University of Naples Federico II, Naples, Italy. [4]Institute of Molecular Biotechnology of the Austrian Academy of Sciences (IMBA), Vienna BioCenter (VBC), Vienna 1030, Austria. [5]Ocular Angiogenesis Group, Department of Ophthalmology, Amsterdam UMC, Meibergdreef 15, Amsterdam 1105 AZ, The Netherlands. [6]Regenerative Medicine Program, Ottawa Hospital Research Institute, Ottawa, Canada. [7]Department of Rheumatology and Clinical Immunology, University Medical Center Groningen, University of Groningen, Groningen 9713 GZ, The Netherlands. [8]Present address: Division of Hematology, Department of Medicine, Stanford University School of Medicine, Stanford, USA. ✉E-mail: d.cinat@stanford.edu; l.barazzuol@umcg.nl; r.p.coppes@umcg.nl

Cinat et al, 2023, 2025); however, the impact of irradiation on specific stem/progenitor populations important for regeneration is still unknown.

The Notch signaling pathway is one of the key regulators of stem cell fate decision and self-renewal capacity in several tissues (Siebel and Lendahl, 2017). The canonical Notch signaling cascade consists of a highly conserved cell–cell communication mechanism, triggered by the interaction between the ligand expressed on one cell, and the corresponding Notch receptor on a neighboring cell (Zhou et al, 2022; Siebel and Lendahl, 2017). Subsequent to ligand–receptor binding, cleavage of the intracellular domain of the Notch (NICD) receptor by γ-secretase allows NCID translocation into the nucleus and transcription of its target genes (Zhou et al, 2022; Siebel and Lendahl, 2017). Although the Notch pathway has been extensively studied in various tissues and organs (Chatzeli et al, 2023; Remark et al, 2023; Pajcini et al, 2011; Gioftsidi et al, 2022), its role in regulating glandular stem/progenitor cells remains to be fully elucidated.

In this study, we employed single-cell RNA sequencing (scRNA-seq), bulk RNA and ATAC-sequencing to explore the dynamic development of a mouse salivary gland organoid (mSGO) model across two different temporal stages, as well as following photon and proton irradiation, two clinically relevant radiation types for the treatment of head and neck cancer. We identified novel adult stem/progenitor features of *Sox9*- and *Itgb1/Cd44*-expressing cells, highlighting their primitive characteristics relative to the other populations. Furthermore, we revealed for the first time that Notch signaling plays a crucial role in regulating the regenerative capacity of both mouse and human submandibular salivary gland organoid-derived stem/progenitor cells after radiation damage, as well as in human mammary and thyroid gland organoids.

# Results

## scRNA-seq of mSGOs show distinct populations resembling salivary gland tissue composition

To obtain a deeper understanding of the populations constituting salivary gland organoids, adult mouse submandibular salivary gland tissue was dissociated, and single cells were placed and cultured in Matrigel (Rocchi et al, 2021b). After one passage, mSGOs were collected on days 7 and 11 of culturing and processed for scRNA-seq and subsequent analysis (Fig. 1A). These time points were chosen as they represent the optimal stages for obtaining fully formed salivary gland organoids (Nanduri et al, 2014; Maimets et al, 2016) for studying the regenerative response of salivary gland stem/progenitor cells in vitro (Rocchi et al, 2022; Cinat et al, 2025). Upon filtering and exclusion of low-quality cells (see "Methods"), 7510 cells from 7-day (Fig. 1B) and 11,248 cells from 11-day organoids (Fig. 1C) were mapped. Unsupervised clustering was performed, resulting in 6 distinct clusters for 7-day organoids (Fig. 1D) and 8 distinct clusters for 11-day organoids (Fig. 1E). Based on established markers (Hauser et al, 2020; Maimets et al, 2016; Song et al, 2018) we identified two populations of potential stem/progenitor cells, namely: stem/progenitor cell I (*Cd164, Sox9, Cd24a*) and stem/progenitor cells II (*Itgb1, Cd44, Trp63*); basal duct cells (*Krt14, Krt5*); luminal duct cells (*Krt8, Krt7/17, Cldn4, Clic1*); and, cycling cells (*Mki67, Top2a*) in both 7-day

and 11-day mSGOs (Fig. 1D–G; Datasets EV1 and EV2). Additionally, using cell type-specific gene signatures extrapolated from the salivary gland tissue dataset GSE150327 (Hauser et al, 2020), we confirmed the identity of the luminal, basal, pro-acinar, and myoepithelial-like clusters, which closely aligned with their respective tissue-specific signature genes (Appendix Fig. S1A). Interestingly, stem/progenitor I and II populations showed strong correlation with intercalated ducts and *Ascl3+* ducts, respectively (Appendix Fig. S1A), regions previously proposed to harbor stem/progenitor populations (Rocchi et al, 2021a; Rugel-Stahl et al, 2012).

7-day mSGOs were mostly composed of basal duct and stem/progenitor cells (Appendix Fig. S1B), which constitute the most plastic cells in the salivary gland (Rocchi et al, 2021a), indicative of a premature phenotype. In contrast, analysis of 11-day mSGOs revealed a shift towards a more differentiated phenotype including an increased number of luminal duct cells, pro-acinar cells (*Chrm3, Alcam, Clcn3*) (Yoon et al, 2022; Maria et al, 2012; Arreola et al, 2002) and myoepithelial-like cells (*Trp63, Lgals7, Csrp1, Sparc*) (Demers et al, 2010; Mauduit et al, 2024; Min et al, 2020) (Fig. 1E, G; Appendix Fig. S1C), aligning with the findings of our previously published study and closely resembling the in vivo tissue composition (Maimets et al, 2016). Salivary gland stem/progenitor cells are believed to reside primarily within the ductal compartment of adult salivary glands (Van Luijk et al, 2015). To test this hypothesis and compare the expression profiles of mSGOs with those of adult tissue, we performed scRNA-seq analysis of freshly isolated murine submandibular salivary glands and compared them with the 7-day mSGO dataset. After filtering out low-quality cells, 2519 cells from the salivary gland tissue samples were mapped (Appendix Fig. S1D,E). Interestingly, a subpopulation of duct cells extrapolated from the scRNA-seq tissue dataset (Fig. 1H; Appendix Fig. S1D) showed a strong expression of genes associated with both mSGO-stem/progenitor cells, such as *Cd164, Sox9, Cd24a, Itgb1* and *Cd44* (Fig. 1I; Appendix Fig. S1F). Further scRNA-seq analysis of enriched salivary gland *Epcam+* cells from E-MTAB-13374 (McKendrick et al, 2023) (Appendix Fig. S1G), identified two distinct populations characterized by elevated expression of genes associated with the organoid stem/progenitor I and stem/progenitor II populations (Fig. 1J).

To further assess the similarity between mSGOs and adult salivary gland tissue, we integrated the mSGOs datasets with our salivary gland tissue dataset and the one from E-MTAB-13374 (McKendrick et al, 2023) (Appendix Fig. S2A). As expected, although organoid populations clustered closely with their respective tissue populations, mSGOs showed enrichment for progenitor-associated markers, such as *Epcam, Cd24a, Krt6a* and *Krt14*, and lacked more differentiated cell types, including acinar, macrophages and endothelial populations (Appendix Fig. S2A,B).

Overall, these data confirm an enrichment of salivary gland stem/progenitor cells within younger mSGOs and the presence of more differentiated cell types in older mSGOs. Nevertheless, mSGOs retain a more progenitor-like phenotype compared to native tissue. Additionally, these findings reiterate the presence of potential salivary gland stem cell-like populations within the ductal compartment of the salivary glands.

In line with the scRNA-seq data, immunofluorescence staining of organoid sections showed the presence of two distinct populations of duct cells: luminal duct cells identified by KRT8/

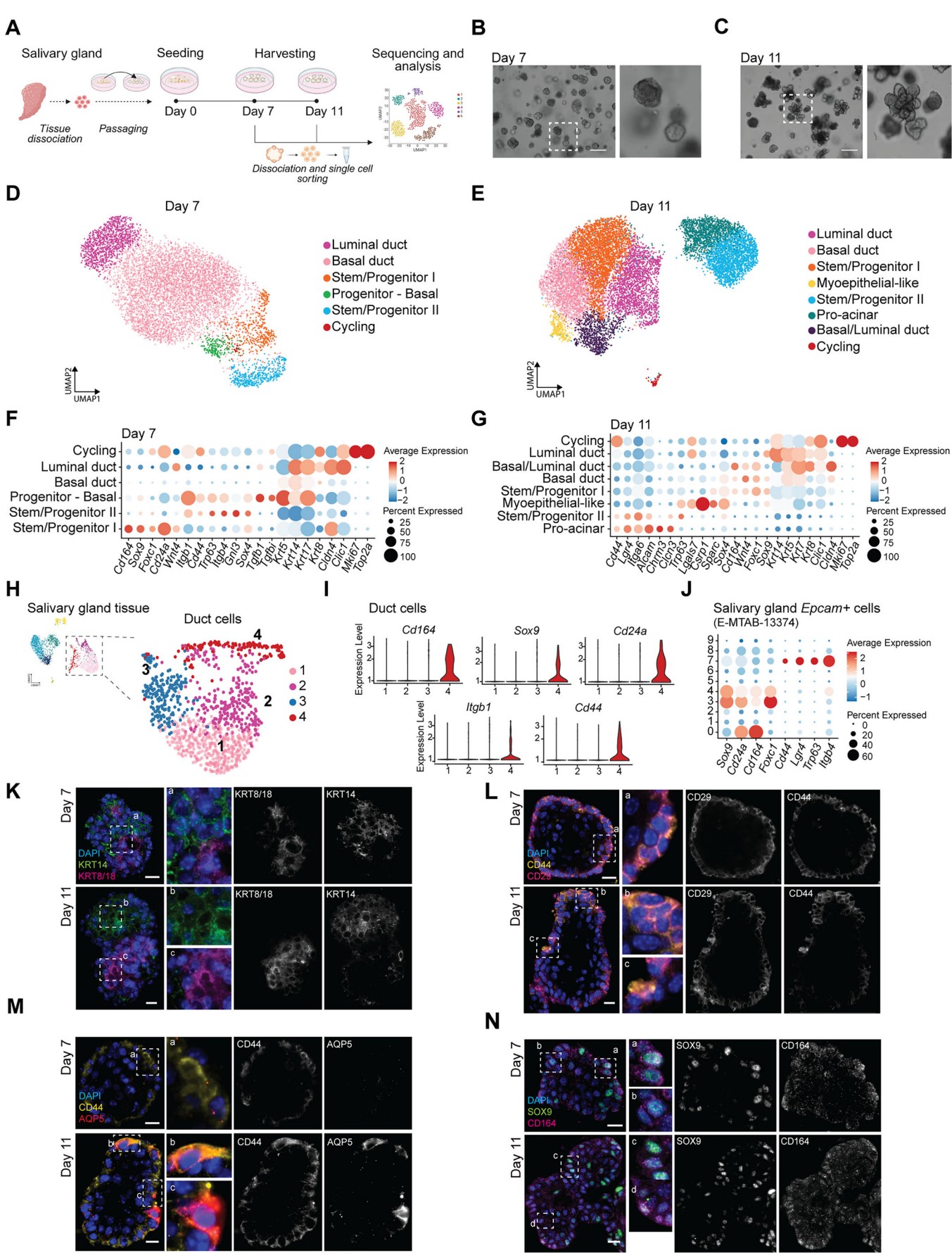

**Figure 1.  scRNA-seq of mSGOs reveals distinct cell populations and an enrichment of stem/progenitor cells.**

(A) Schematic representation of the scRNA-seq experiment. (B) Representative image of 7-day mSGOs. Scale bar, 100 μm. (C) Representative image of 11-day mSGOs. Scale bar, 100 μm. (D) UMAP of the 7-day mSGO dataset showing the main cell populations. (E) UMAP of the 11-day mSGO dataset showing the main cell populations. (F) Dot plot showing cell type marker genes of the 7-day mSGO dataset. (G) Dot plot showing representative marker genes of the 11-day mSGO dataset. (H) UMAP of a subpopulation of duct cells extrapolated from the scRNA-seq dataset of salivary gland tissue showing the main cell sub-clusters after re-clustering. The entire UMAP of salivary gland tissue is shown in Appendix Fig. S1D. (I) Violin plots generated from scRNA-seq data showing the expression of *Cd164, Sox9, Cd24a, Itgb1* and *Cd44* across the different Duct sub-clusters. Colors are the same as in (H). (J) Dot plot showing mSGO stem/progenitor cell markers in salivary gland *Epcam+* cell from E-MTAB-13374 (McKendrick et al, 2023). Cluster numbers represent different populations and are the same as in Appendix Fig. S1G. (K) Representative images of immunofluorescence staining of 7-day and 11-day mSGOs showing the expression of KRT14 and KRT8/18. Scale bar, 10 μm. (L) Representative images of immunofluorescence staining of 7-day and 11-day mSGOs showing the expression of CD29 and CD44. Scale bar, 10 μm. (M) Representative images of immunofluorescence staining of 7-day and 11-day mSGOs showing the expression of CD44 and AQP5. Scale bar, 10 μm. (N) Representative images of immunofluorescence staining of 7-day and 11-day mSGOs showing the expression of SOX9 and CD164. Scale bar, 10 μm. Source data are available online for this figure.

18 and basal duct cells identified by KRT14 (Fig. 1K). Moreover, we identified cells positive for CD29 (*Itgb1*) and CD44, constituting a population of stem/progenitor cells (Fig. 1L). Interestingly, AQP5, a well-known pro-acinar and acinar marker, was found to be expressed by some CD44-positive cells, especially in 11-day mSGOs (Fig. 1M). This aligned with our scRNA-seq data (Fig. 1E), which suggested a potential role for the *Itgb1/Cd44*-expressing stem/progenitor population as acinar precursor cells. *Cd164*, a gene encoding a sielomucin protein described as a novel human stem cell marker (Watt et al, 2021), was highly expressed by the *Sox9+* stem/progenitor cluster (Fig. 1F,G). Immunofluorescence staining showed a more ubiquitous expression of CD164 compared to scRNA-seq data; however, several SOX9-expressing cells exhibited notably higher CD164 levels than other cells (Fig. 1N). Together, our data establish the existence of distinct cell populations within our mSGOs and provide new insights into salivary gland stem/progenitor cell populations and their markers.

## Salivary gland stem/progenitor cells are functional in mSGOs

Based on known stem/progenitor cell markers, scRNA-seq analysis revealed the presence of two salivary gland stem/progenitor populations within mSGOs (Fig. 1D,E). Notably, genes identifying these populations were strongly associated with stem cell-related gene ontology terms, such as embryonic and tissue development-related biological processes (Appendix Fig. S3A), indicating stem/progenitor cell features. To confirm these findings, we performed ATAC-sequencing (ATAC-seq) of 7-day mSGOs, which exhibited a more premature phenotype compared to 11-day mSGOs. This analysis revealed an enrichment of several motifs associated with transcription factors strongly linked to embryonic development, tissue development, differentiation and morphogenesis-related processes (Appendix Fig. S3B,C). Specifically, motifs associated with KLF, TEAD, NF-Y, RUNX, AP2, SOX and FOX, which are known regulators of embryonic and stem cell-related processes (Fu et al, 2021; Sarkar and Hochedlinger, 2013; Jiang et al, 2008; Currey et al, 2021; Rigillo et al, 2021; Kim et al, 2014), were highly enriched in the stem/progenitor clusters, particularly of 7-day mSGOs (Fig. 2A), suggesting heightened stem cell features and supporting the more immature profile of younger mSGOs. To further support this notion, we integrated the 7-day and 11-day mSGO datasets (Fig. 2B,C; Appendix Fig. S3D) and performed unbiased RNA velocity and pseudotime analyses (Fig. 2D,E). Notably, cluster analysis and RNA velocity confirmed spontaneous differentiation

over time, as evidenced by the directionality of the velocity flow (Fig. 2D) and the enrichment of more differentiated markers within the 11-day mSGO populations (Appendix Fig. S3D).

Based on RNA velocity analysis, we identified two precursor populations in the 7-day mSGOs: basal cells and stem/progenitor I cells (Fig. 2D). Pseudotime analysis further revealed that while stem/progenitor I cells appear to give rise to more duct-like cells, basal cells progressed towards the stem/progenitor cell II and pro-acinar populations, suggesting potential plasticity and de-differentiation capacity of these cells in vitro (Fig. 2E). This finding aligns with previous studies (Kwak and Ghazizadeh, 2015; Kwak et al, 2018; May et al, 2018), which report that KRT14[+] basal cells exhibit progenitor-like features and can replace other type of cells after damage.

Together, these data indicate an enrichment of stem/progenitor cells in 7-day mSGOs and highlight *Sox9*, *Krt14-*, and *Itgb1-* expressing cells as the most primitive cell types within our mSGOs.

ScRNA-seq data revealed that *Itgb1* (CD29) was highly upregulated in the stem/progenitor cell cluster II, whereas *Cd24a* (CD24), previously associated with a potential population of salivary gland stem cells (Nanduri et al, 2014), was highly enriched in the stem/progenitor I population (Fig. 1F; Appendix Fig. S4A). To validate the stem cell properties of these putative stem/progenitor cells identified in the mSGO datasets, we used fluorescence-activated cell sorting (FACS). Based on the surface markers highly expressed in our scRNA-seq data, we sorted the stem/progenitor populations from 7-day mSGOs. We next assessed their ability to form organoids and self-renew in vitro (Fig. 2F; Appendix Fig. S4B,C). When sorted cell populations were cultured in Matrigel, CD29[high]/CD24[low] and CD29[low]/CD24[high] expressing-cells (Fig. 2G) demonstrated a significantly greater capacity to form organoids (Fig. 2H,I) and ability to self-renew, as indicated by a higher secondary OFE (Appendix Fig. S4D,E), compared to the CD29[low]/CD24[low]-population, thus confirming their stem cell properties. Additionally, we identified and sorted a CD29[high]/CD24[high] population (Fig. 2G) that also exhibited high self-renewal capacity (Fig. 2I; Appendix Fig. S4E), indicating the presence of a double-positive stem/progenitor population. These findings corroborate the presence and functional capabilities of salivary gland stem/progenitor cells within our mSGOs, substantiating our previous findings (Nanduri et al, 2014).

Further scRNA-seq analysis revealed an upregulation of genes associated with mesenchymal features, including *Cd44, Itgb4, Nt5e* (Kröger et al, 2019), and a set of epithelial/mesenchymal (E/M)-related genes (EMT), in the *Itgb1*-expressing populations (Fig. 2J).

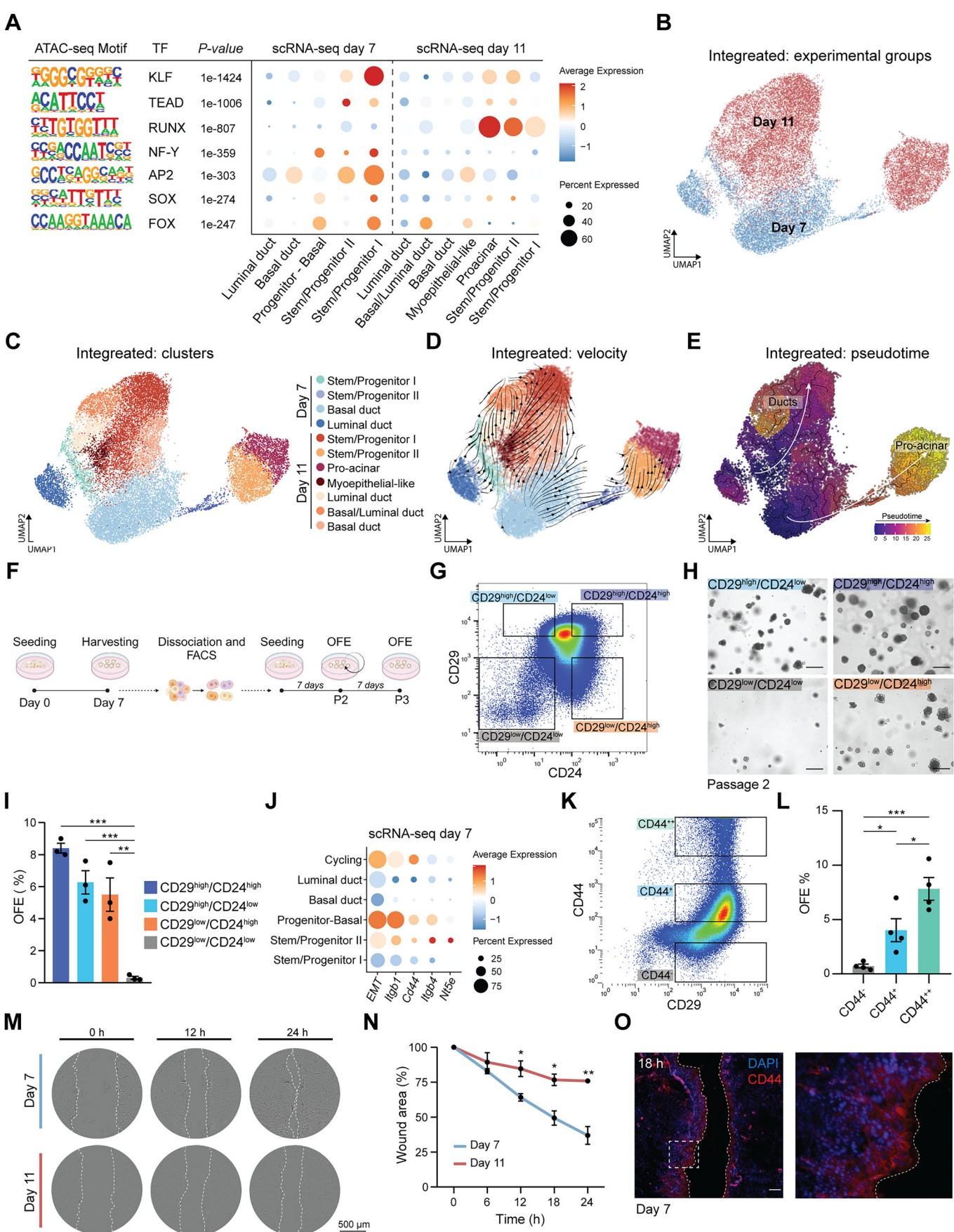

**Figure 2.   *Sox9*- and *Cd44/Itgb1*-expressing cells display stem/progenitor features.**

(A) Table showing the ATAC-seq motifs and relative *p* values of KLF, TEAD, RUNX, NF-Y, AP2, SOX and FOX transcription factor (TF) families. Expression of the genes associated with each motif was extrapolated from the 7-day and 11-day mSGO dataset. (B) UMAP of the 7-day and 11-day mSGO-integrated dataset showing the different conditions. (C) UMAP of the 7-day and 11-day mSGO-integrated dataset showing the main cell populations. Cluster annotation is shown in Appendix Fig. S3D. (D) Velocity plot of the 7-day and 11-day mSGO-integrated dataset showing the projected velocities. Cluster colors match those in (C). (E) Trajectory plot of the 7-day and 11-day mSGO-integrated dataset showing the pseudotime course. White arrows indicate Ducts and Pro-acinar trajectories. (F) Schematic representation of the self-renewal assay following FACS of 7-day mSGOs. (G) Representative FACS plot showing the gating strategy. Unstained and backgating are shown in Appendix Fig. S4B,C. (H) Representative images of sorted cells after 1 week in culture (P2). Scale bar, 100 μm. (I) Organoid quantification of sorted cells after 1 week in culture (P2) shown as organoid formation efficiency (OFE) (means ± s.e.m.; *n* = 3 animals/condition). One-way ANOVA, post-hoc Tukey's test (CD29$^{high}$/CD24$^{high}$ vs. CD29$^{low}$/CD24$^{low}$ ***$p$ = 0.0001; CD29$^{high}$/CD24$^{low}$ vs. CD29$^{low}$/CD24$^{low}$ ***$p$ = 0.0009; CD29$^{low}$/CD24$^{high}$ vs. CD29$^{low}$/CD24$^{low}$ ***$p$ = 0.0022). (J) Dot plot showing mesenchymal-like markers and EMT features (GSEA Mouse Gene Set: Hallmark Epithelial Mesenchymal Transition) in the 7-day mSGO dataset. (K) Representative FACS plot showing the gating strategy. (L) Organoid quantification of sorted cells after 1 week in culture shown as organoid formation efficiency (OFE %) (means ± s.e.m.; *n* = 4 animals/condition). One-way ANOVA, post-hoc Tukey's test (CD44$^{-}$ vs. CD44$^{+}$ **$p$ = 0.049; CD44$^{+}$ vs. CD44$^{++}$ *$p$ = 0.031; CD44$^{-}$ vs. CD44$^{++}$ ***$p$ = 0.0007).
(M) Representative images of salivary gland cells derived from 7-day and 11-day mSGOs at 0, 12 and 24 h after wound generation. Dotted line shows wound borders. (N) Quantification of the wound area over time (means ± s.e.m.; *n* = 3 animals/condition). Data is relative to 0 h (100%). Two-way ANOVA, post-hoc Bonferroni's multiple comparison test (12 h *$p$ = 0.029; 18 h *$p$ = 0.014; 24 h **$p$ = 0.0037). (O) Representative immunofluorescence images of salivary gland cells derived from 7-day mSGOs, 18 h after wound generation, showing CD44 expression. Representative image of salivary glands cells derived from 11-day mSGOs is shown in Appendix Fig. S4H. Scale bar, 100 μm. Source data are available online for this figure.

Particularly, *Cd44* has been described as a key feature in E/M states, playing an essential role in tumor progression by promoting both stemness and migration of cancer stem cells (Mani et al, 2008), and serving as an important marker for hematopoietic stem cells (Williams et al, 2013). Given its high expression in the stem/progenitor II and cycling populations (Fig. 1F,G), we investigated whether *Cd44*-expressing cells exhibited enhanced stem/progenitor cell properties and increased migratory potential. FACS analysis of CD29$^{high}$ cells showed different levels of CD44 (Fig. 2K), with higher CD44 expression (CD44$^{++}$) correlating with increased OFE, suggesting enhanced stemness features (Fig. 2L). To assess the potential migratory behavior of these cells, we conducted a wound-healing assay in 2D salivary gland cultures. To minimize the influence of proliferation, cultures were serum-starved, as confirmed by reduced KI67 staining (Appendix Fig. S4F). Notably, these salivary gland cultures, containing the primary cell populations found in the 3D organoid model (Appendix Fig. S4G), exhibited a clear migratory potential and the ability to fill the wound over time (Fig. 2M,N). However, salivary gland cells from 11-day mSGOs exhibited a significantly reduced migration compared to those from 7-day mSGOs (Fig. 2N), likely due to their more differentiated phenotype, as evidenced by the loss of stem cell features and upregulation of pro-acinar markers (Fig. 2A; Appendix Fig. S3D). Moreover, CD44-expressing cells accumulated prominently at the wound's migratory front, migrating from the outer regions towards the wound center (Fig. 2O; Appendix Fig. S4H).

These findings, along with the co-expression of AQP5 by some CD44-expressing cells in 11-day mSGOs (Fig. 1M), suggest that CD44 may identify a previously unrecognized salivary gland cell population with stemness, differentiation and migratory properties.

## Irradiation-surviving stem/progenitor cells show increased activity and Notch signaling

To gain a deeper understanding of the effects of radiation on the regeneration capabilities of stem/progenitor populations within our mSGOs, organoids were irradiated with photon and proton, two clinically relevant radiation modalities, followed by scRNA-seq analysis. mSGOs were irradiated at day 5 with 7 Gy photon or proton, a dose previously observed to induce a significant cytotoxic and regenerative response, as well as mechanistic changes consistent with in vivo models (Peng et al, 2020; Cinat et al, 2025), and collected at day 7 for scRNA-seq and analysis (Fig. 3A). Upon exclusion of low-quality cells (see methods), the datasets from control, photon- and proton-irradiated samples were merged (Fig. 3B) and unsupervised clustering was performed (Fig. 3C; Appendix Fig. S5B). Cluster annotation was executed by using the previously identified markers (Fig. 1F), and the presence of similar cell types compared to control samples was identified (Appendix Fig. S5C,D). To detect gene expression differences between control, photon and proton irradiation, we performed differential gene expression (DEG) analysis and gene set enrichment analysis (GSEA) for each cluster individually. In line with our previous works (Cinat et al, 2023, 2025), GSEA analysis revealed an overall upregulation of immune and mitochondrial stress-related gene ontology (GO) terms after irradiation with both photons and protons (Appendix Fig. S5E). However, the stem/progenitor and basal duct cells showed an enrichment in processes related to proliferation, development and tissue morphogenesis. These processes include GO terms such as gland development, tube morphogenesis, tube development and cell proliferation (Appendix Fig. S5E), suggesting an upregulation of stem cell and regenerative features after irradiation. Notably, *Hmga2* and genes members of the Notch signaling pathway, such as *Notch1* and *Hes1*, emerged as the most enriched genes associated with these stem cell-related biological processes (Fig. 3D), suggesting their potential role in the maintenance of the surviving stem/progenitor cells after irradiation.

To investigate potential signaling pathways important for stem/progenitor cell regulation in both control and irradiated samples, we performed a cell–cell interaction analysis using the 7-day control and irradiated mSGO datasets. Given the minimal differences in cell type composition observed between photon and proton irradiation, these samples were analyzed together and hereinafter referred to as the irradiation group. Notch signaling, mediated by *Notch1-Jag1/Jag2* interaction, appeared as one of the most enriched pathways in control mSGOs (Fig. 3E). Intriguingly, irradiation led to an overall heightened communication compared to control samples, as shown by the increased communication

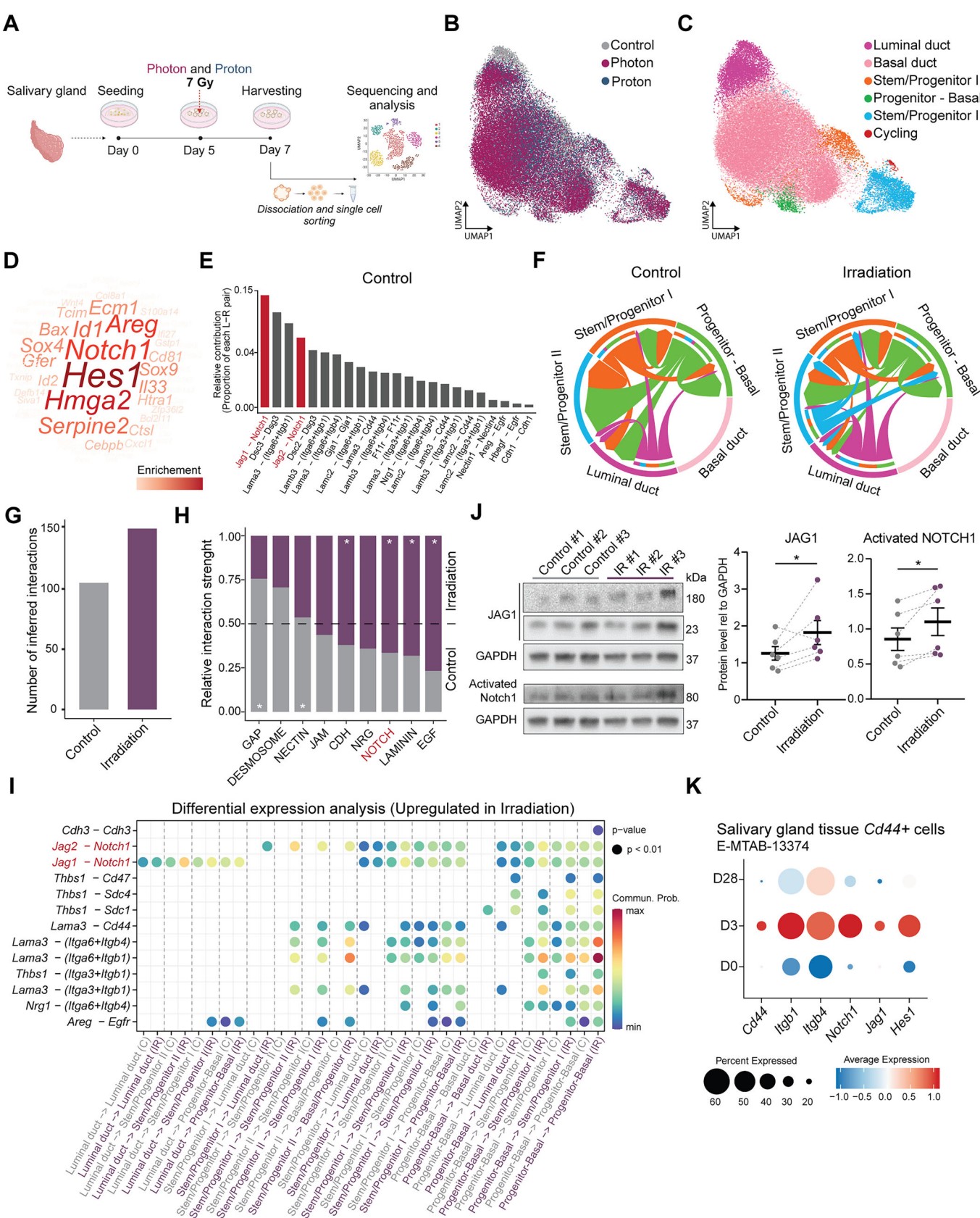

◀

**Figure 3. Surviving stem/progenitor cells show increased communication and Notch signaling after irradiation.**

(A) Schematic representation of the scRNA-seq experiment of photon and proton-irradiated mSGOs. (B) UMAP of control, photon and proton merged datasets showing the different conditions. (C) UMAP of control, photon and proton merged datasets showing the main cell populations. (D) Word cloud plot showing the most enriched genes in developmental, tissue morphogenesis and proliferation-related processes presented in Appendix Fig. S5E. (E) Bar plot showing the proportions of significant ligand–receptor interaction in control mSGOs. Notch-related genes are shown in red. (F) Chord plot showing the interaction strength within each cell cluster in control and irradiated (photon + proton) mSGOs. (G) Bar plot showing the total number of interactions in control and irradiation. (H) Bar plot showing the most enriched signaling pathways shared between control and irradiated mSGOs. Paired Wilcoxon test was used for statistical analysis and *Symbol indicates significant signaling pathways in control or irradiation. (I) Bubble plot showing significantly upregulated ligand–receptor pairs between different cell types after irradiation. The first column always represents control, and the second column represents irradiation. Statistical significance was assessed using the paired Wilcoxon test. (J) Western blot analysis of JAG1, activated NOTCH1 and GAPDH in control and irradiated mSGOs at day 7. Additional biological replicates are shown in Appendix Fig. S6F. Data are normalized to GAPDH (means ± s.e.m.; $n = 6$ animals/condition). JAG1 quantification includes both the 180 kDa and 23 kDa bands. Two-sided paired $t$ test (JAG1 *$p = 0.0408$; Activated NOTCH1 *$p = 0.0209$). (K) Dot plot showing the expression of *Cd44*, *Itgb1*, *Itgb4*, *Notch1*, *Jag1* and *Hes1* in a subpopulation of *Cd44*+ salivary gland cells at day 0 (D0), day 3 (D3) and day 28 (D8) post-irradiation from E-MTAB-13374 (McKendrick et al, 2023). Source data are available online for this figure.

patterns and strength (Fig. 3F,G; Appendix Fig. S6B). Moreover, we observed a significantly higher enrichment of Notch signaling after irradiation (Fig. 3H), particularly within the stem/progenitor populations, which displayed increased outgoing and incoming signaling communication patterns (Appendix Fig. S6C). Differential expression analysis of ligand–receptor interactions using CellChat confirmed the upregulation of several signaling pathways following irradiation, including Notch signaling, which was mediated by *Notch1-Jag1* and *Notch1-Jag2* interactions (Fig. 3I). Furthermore, compared to other ligand–receptor pairs, Notch-related genes appeared more broadly expressed among the various cell populations (Fig. 3I; Appendix Fig. S6D), further supporting a key role for Notch signaling in maintaining cell function under regenerative conditions. Western blot analysis, showing a significant upregulation of activated NOTCH1 and JAG1 2 days post-irradiation (Fig. 3J; Appendix Fig. S6F), together with an overall higher expression of *Hes1* detected in irradiated samples by scRNA-seq analysis (Appendix Fig. S6G), further confirmed these findings at the protein and gene expression level, respectively. Of note, proton irradiation resulted in slightly higher expression of Notch-related genes and more pronounced communication patterns compared to photon irradiation (Appendix Fig. S6C,E). scRNA-seq analysis of salivary gland tissue from E-MTAB-13374 (McKendrick et al, 2023) confirmed these findings in an in vivo setting. *Cd44*-expressing cells showed a significant enrichment of Notch signaling-related genes (Appendix Fig. S6H), with a marked increase of *Notch1*, *Jag1*, and *Hes1* observed 3 days post-irradiation (Fig. 3K).

Taken together, these results indicate a potential role for Notch signaling in regulating stem/progenitor cell activity after radiation damage.

## Salivary gland stem/progenitor cells rely on Notch signaling to maintain their self-renewal and migratory capacity

Notch signaling plays a pivotal role in the regulation of stem cell activity and regeneration in several tissues (Siebel and Lendahl, 2017). Since Notch appeared as one of the most enriched signaling pathways in both control and irradiated mSGOs (Fig. 3E,I), we investigated its role in maintaining the self-renewal capacity of salivary gland stem/progenitor cells in vitro. The self-renewal capacity of non-irradiated mSGOs, cultured in medium without Wnt and R-Spondin, factors that interact with Notch signaling and

allow increased OFE (Nanduri et al, 2014), was assessed upon treatment with the Notch γ-Secretase Inhibitor Dibenzazepine (DBZ) (Fig. 4A). Notably, Notch inhibition (Appendix Fig. S7A) impaired mSGO growth (Fig. 4B) and significantly decreased the self-renewal capacity of salivary gland stem/progenitor cells, shown as a reduction of OFE and population doubling (PD) (Fig. 4C). Moreover, DBZ-treated mSGOs showed increased expression of acinar markers *Prol1* and *Aqp5*, suggesting premature differentiation toward an acinar-like phenotype (Appendix Fig. S7B). Next, to assess whether Notch inhibition influenced the differentiation capacity and stem cell exhaustion of mSGOs, untreated 7-day mSGOs were placed in differentiation media with or without DBZ, and their ability to differentiate and self-renew was assessed (Fig. 4D). In line with the previous findings, untreated organoids (−DBZ) showed a more spherical morphology (Fig. 4E, left panel; Appendix Fig. S7C) and high levels of the basal/progenitor marker KRT14 (Appendix Fig. S7D), indicative of a more premature and undifferentiated phenotype. In contrast, treatment with DBZ (+DBZ) or the Notch γ-Secretase Inhibitor DAPT (+DAPT) (Fig. 4E, right panel; Appendix Fig. S7C) led to the formation of differentiated mini gland-like structures and the increased expression of the acinar marker AQP5 (Appendix Fig. S7D). This aligned with previous work (Yoon et al, 2022), where Notch inhibition led to terminal differentiation and organoid maturation. Furthermore, mini glands treated with DBZ showed a much lower self-renewal capacity compared to untreated organoids (Fig. 4F,G), confirming stem/progenitor cell exhaustion. To assess the impact of DBZ treatment on cell migration, we conducted a wound healing assay as previously described (Fig. 2M,N). Notch inhibition reduced the motility of CD44-expressing cells (Fig. 4H; Appendix Fig. S7E), with DBZ-treated samples showing a significantly lower capacity to fill the wound area compared to DMSO-treated samples (Fig. 4I). Taken together, these data indicate that salivary gland stem/progenitor cells need Notch signaling to maintain their stemness, prevent the formation of differentiated structures, and sustain their motility in control conditions.

After irradiation, Notch signaling was found to be highly upregulated (Fig. 3I; Appendix Fig. S7F,G). To address if surviving stem/progenitor cells rely on Notch to maintain their self-renewal capacity after irradiation, mSGOs were irradiated at day 5 and treated with either the Notch inhibitor DBZ or the Notch activator JAG1 from day 7. Treated 11-day mSGOs were collected, and self-renewal capacity was assessed (Fig. 4J). It is important to note that in this assay, organoids were treated when fully formed, rather than

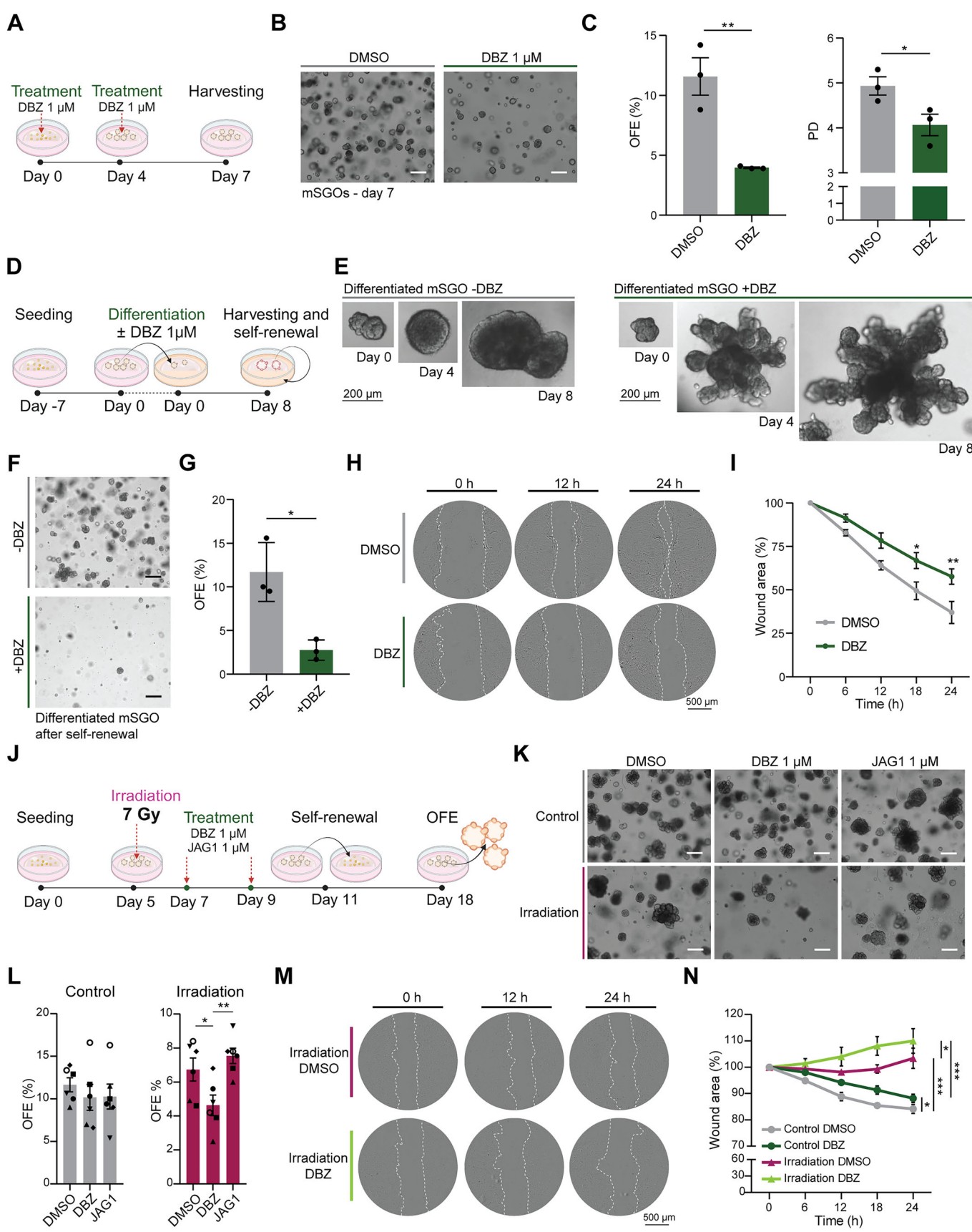

**Figure 4. mSGOs rely on Notch signaling to maintain stemness and prevent premature differentiation.**

(A) Schematic representation of DBZ treatment in control mSGOs. (B) Representative images of 7-day mSGOs after treatment with DMSO or DBZ. Scale bar, 100 μm. (C) Organoid quantification of DMSO and DBZ-treated mSGOs shown as OFE% and population doubling (PD) (means ± s.e.m.; $n = 3$ animals/condition). Two-sided unpaired $t$ test (OFE **$p = 0.0082$; PD *$p = 0.0501$). (D) Schematic representation of mSGO differentiation assay with or without DBZ treatment. (E) Representative images of differentiated mSGOs with or without DBZ treatment. Scale bar, 200 μm. Whole-well images are shown in Appendix Fig. S7C. (F) Representative images of differentiated mSGOs treated with or without DBZ after self-renewal. Scale bar, 100 μm. (G) Organoid quantification of differentiated mSGOs treated with or without DBZ after self-renewal shown as OFE% (means ± s.e.m.; $n = 3$ animals/condition). Two-sided unpaired $t$ test (OFE *$p = 0.014$). (H) Representative images of salivary gland cells after DMSO or DBZ treatment at 0, 12 and 24 h after wound generation. Dotted line shows wound borders. DMSO panels are the same as Day 7 in Fig. 2M. (I) Quantification of the wound area over time. Data is relative to 0 h (100%) (means ± s.e.m.; $n = 3$ animals/condition). Two-way ANOVA, post-hoc Sidak's multiple comparison test (12 h $p = 0.0714$; 18 h *$p = 0.0190$; 24 h **$p = 0.0047$). (J) Schematic representation of DBZ and JAG1 treatment in control and photon-irradiated mSGOs. (K) Representative images of control and photon-irradiated mSGOs after treatment with DMSO, DBZ or JAG1. Scale bar, 100 μm. (L) Organoid quantification of control and photon-irradiated mSGOs after treatment with DMSO, DBZ or JAG1 shown as OFE (means ± s.e.m.; $n = 5$ animals/condition). Different dot shapes indicate different biological replicates. One-way repeated measures ANOVA, post-hoc Tukey's test (DMSO vs. DBZ *$p = 0.035$; DBZ vs. JAG1 **$p = 0.0058$). (M) Representative images of irradiated salivary gland cells after DMSO or DBZ treatment at 0, 12 and 24 h after wound generation. Dotted line shows wound borders. (N) Quantification of the wound area over time. Data is relative to 0 h (100%) (means ± s.e.m.; $n = 3$ animals/condition). Two-way ANOVA, post-hoc Tukey's test (Control DMSO vs. Control DBZ *$p = 0.045$; Control DMSO vs. Irradiation DMSO ***$p = 0.0003$; Irradiation DMSO vs. Irradiation DBZ *$p = 0.049$; Control DBZ vs. Irradiation DBZ ***$p = 0.0001$). Source data are available online for this figure.

as single cells as done in non-irradiated conditions (Fig. 4A). Interestingly, DBZ treatment impaired the self-renewal capacity of irradiated mSGOs, shown as by a reduction in OFE (Fig. 4K,L; Appendix Fig. S7H). Additionally, Notch inhibition significantly impaired the migratory abilities of irradiated salivary gland cells, which were already reduced compared to non-irradiated samples (Fig. 4M,N), highlighting the critical role of Notch signaling in sustaining both stemness and migratory potential in irradiated salivary gland tissue. Notably, while JAG1 treatment showed a trend toward increased OFE in photon-irradiated samples (Fig. 4K,L), no changes were detected in proton-irradiated mSGOs (Appendix Fig. S7H), which already exhibited higher levels of Notch signaling at the single-cell level (Appendix Fig. S6E).

## Notch signaling maintains the stem cell potential of human salivary, thyroid and mammary gland-derived organoids

To extend our findings to other glandular tissues, we employed our previously described mouse thyroid gland organoid (mTGO) model (Ogundipe et al, 2021; van der Vaart et al, 2021) and assessed the role of Notch signaling in regulating mTGO self-renewal capacity and differentiation in vitro. Similar to mSGOs, DBZ treatment led to downregulation of Notch-related genes (Appendix Fig. S8A) and to a significant reduction of the self-renewal capacity of non-irradiated mTGOs (Fig. 5A,B). Furthermore, DBZ-treated organoids showed increased lobular morphology (Fig. 5A,C) and heightened expression of the thyroid differentiation markers *Slc5a5* (NIS) and *Tg* (Thyroglobulin) (Appendix Fig. S8B), suggesting premature differentiation. To assess the differentiation capacity of mTGOs upon inhibition of Notch signaling, we placed untreated 7-day mTGOs in differentiation media (Ogundipe et al, 2021) with or without DBZ. Similar to mSGOs (Fig. 4E), DBZ-treated mTGOs showed a lobular phenotype (Fig. 5D) accompanied by the upregulation of differentiation markers (Appendix Fig. S8C).

Despite the absence of a thyroid-specific stem cell marker, our previous work identified the expression of several stem cell-related markers in mTGOs, including CD29 (*Itgb1*) and CD44 (Ogundipe et al, 2021). To determine whether these cells possess organoid-forming capacity like those in mSGOs, we assessed their self-

renewal potential following FACS analysis (Fig. 5E; Appendix Fig. S8D). Similar to mSGO, higher CD44 expression correlated with increased OFE (Fig. 5F; Appendix Fig. S8E), with comparable OFE between the CD44$^{++}$ and CD44$^{+}$ populations (Fig. 5F). To investigate the migratory properties of this potential stem/progenitor population, we serum-starved thyroid cells (Appendix Fig. S8F) and performed a wound healing assay. Already 24 h after wound generation, thyroid cells showed a pronounced migratory phenotype, which was further enhanced by Notch signaling activation with JAG1 (Fig. 5G; Appendix Fig. S8G). Interestingly, as observed in salivary gland cells, CD44-expressing cells were predominantly localized at the wound front (Fig. 5H). Together, these findings emphasize the role of Notch signaling in mTGO self-renewal and differentiation and reveal a CD44-expressing cell population with stemness and migratory abilities.

A more profound comprehension of the signaling pathways governing patient-derived stem/progenitor cells can help improve the use and establishment of organoid models for preclinical studies and regenerative applications (Soto-Gamez et al, 2024). To translate our findings to patient-derived organoid models, we cultured organoids using biopsies from patients' salivary glands. We assessed how Notch signaling influenced organoid growth and self-renewal capacity in vitro (Appendix Fig. S9A). Similar to mSGOs, human submandibular salivary gland organoids (hSGOs) expressed basal and stem/progenitor cell markers (Fig. 5I) and a significant decrease in self-renewal capacity following DBZ treatment (Fig. 5J,K; Appendix Fig. S9B). To assess the regenerative capacity of human salivary gland stem/progenitor cells after irradiation, we treated irradiated hSGOs with DBZ and assessed their self-renewal capacity as previously described (Fig. 4J). Notably, inhibition of Notch signaling after irradiation markedly impaired organoid growth, as evidenced by the significant decrease in PD after self-renewal (Fig. 5L; Appendix Fig. S9C). These findings confirmed the conserved role of Notch signaling in regulating both murine and patient-derived SGOs in regenerative conditions. To extend these observations to other human glandular tissues, we generated organoids from patient biopsies of human thyroid (hTGOs) and mammary glands (hMGOs) (Appendix Fig. S9A). Interestingly, all human glandular organoids exhibited an impairment in self-renewal and a significantly lower OFE upon DBZ treatment (Fig. 5M–P; Appendix Fig. S9D,E), highlighting a

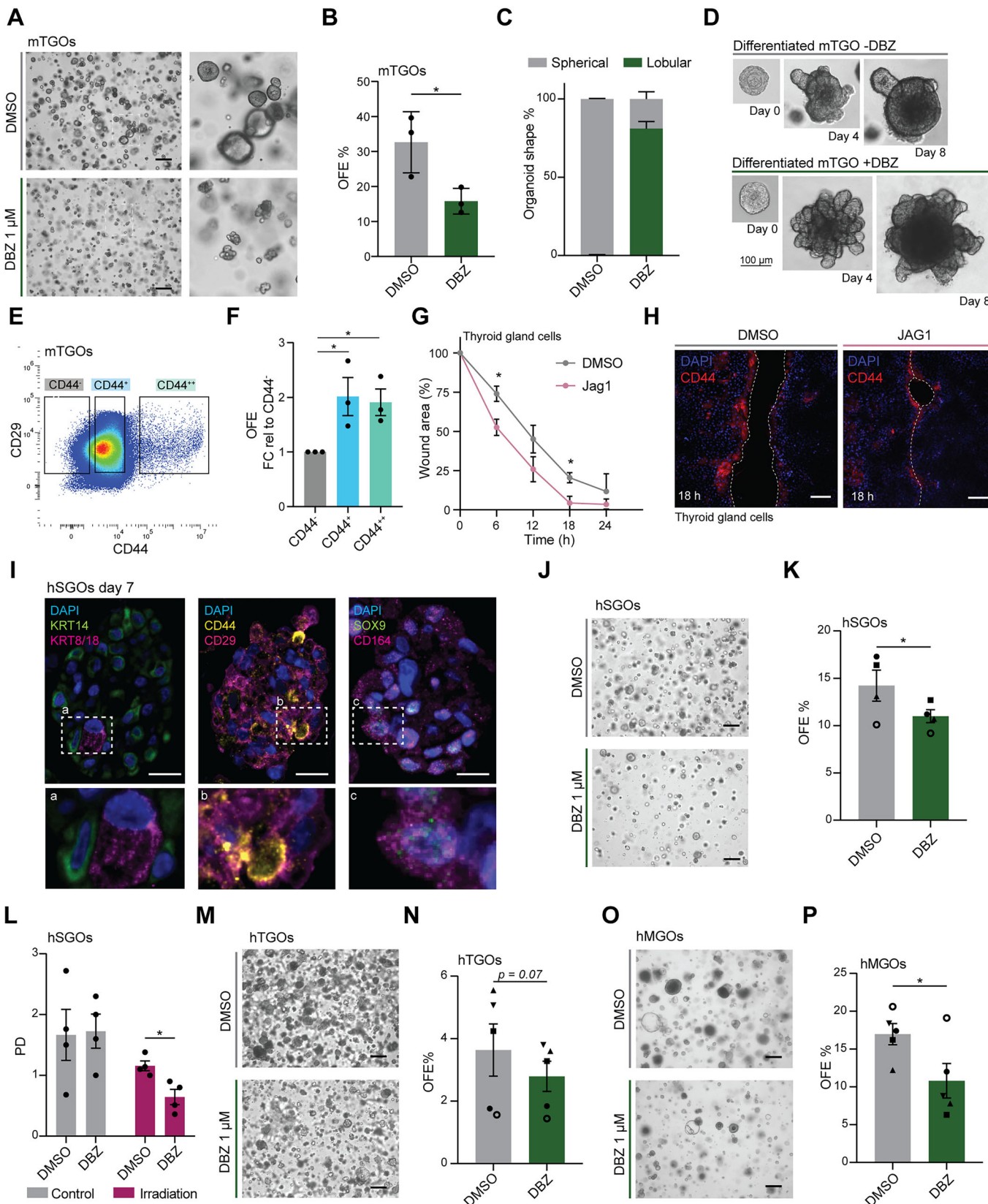

**Figure 5. Notch signaling regulates stem cell self-renewal and differentiation of murine and patient-derived glandular organoids.**

(A) Representative images of 7-day-old mTGOs after treatment with DMSO or DBZ. Scale bar, 100 μm. (B) Organoid quantification of DMSO and DBZ-treated mTGOs shown as OFE% (means ± s.e.m.; $n = 3$ animals/condition). Two-sided unpaired $t$ test (OFE $*p = 0.045$). (C) Quantification of lobular and spherical mTGOs after DMSO and DBZ treatment. Data is shown as percentage (%) of lobular and spherical organoids/total organoid number (means ± s.e.m.; $n = 3$ animals/condition). (D) Representative images of differentiated mTGOs with or without DBZ treatment. Scale bar, 100 μm. (E) Representative FACS plot showing the gating strategy. Backgating is shown in Appendix Fig. S8D. (F) Organoid quantification of sorted cells after 1 week in culture shown as organoid formation efficiency (OFE). Data is shown as FC relative to CD44$^-$ (means ± s.e.m.; $n = 3$ animals/condition). Two-sided unpaired $t$ test (CD44$^-$ vs. CD44$^+$ $*p = 0.044$; CD44$^{++}$ vs. CD44$^-$ $**p = 0.019$). (G) Quantification of the wound area over time. Data is relative to 0 h (100%) (means ± s.e.m.; $n = 3$ animals/condition). One-way ANOVA, post-hoc Tukey's test (6 h $*p = 0.046$; 18 h $*p = 0.05$). (H) Representative images of immunofluorescence staining of thyroid glands cells 24 h after wound generation and DMSO/JAG1 treatment showing the expression of CD44. Scale bar, 100 μm. (I) Representative images of immunofluorescence staining of hSGOs showing the expression of KRT14, KRT8/18, CD29, CD44, SOX9 and CD164. Scale bar, 10 μm. (J) Representative images of hSGOs after DMSO and DBZ treatment. Scale bar, 100 μm. (K) hSGO quantification shown as OFE after DMSO and DBZ treatment (means ± s.e.m.; $n = 4$ patients/condition). Different dot shapes indicate different patients. Two-sided paired $t$ test ($*p = 0.049$). (L) Organoid quantification of control and photon-irradiated hSGOs after treatment with DMSO and DBZ shown as PD (means ± s.e.m.; $n = 4$ patients/condition). Two-sided paired $t$ test ($*p = 0.014$). (M) Representative images of hTGOs after DMSO and DBZ treatment. Scale bar, 100 μm. (N) hTGO quantification shown as OFE after DMSO and DBZ treatment (means ± s.e.m.; $n = 5$ patients/condition). Different dot shapes indicate different patients. Two-sided paired $t$ test. (O) Representative images of hMGOs after DMSO and DBZ treatment. Scale bar, 100 μm. (P) hMGO quantification shown as OFE after DMSO and DBZ treatment (means ± s.e.m.; $n = 5$ patients/condition). Different dot shapes indicate different patients. Two-sided paired $t$ test ($*p = 0.026$). Source data are available online for this figure.

conserved pre-regenerative role of Notch signaling across various patient-derived glandular organoid models.

## Discussion

Salivary gland stem/progenitor cells are a rare population of cells essential for normal tissue turnover in regenerative conditions. The establishment of SGOs has greatly advanced the characterization of these cells. Yet, the markers and the mechanisms defining them remain to be fully elucidated. This is the first study in which scRNA-seq profiling was used to characterize a mSGO model across different timepoints before and after irradiation. Bioinformatic and functional analyses identified *Sox9*, *Itgb1* and *Cd44* as novel markers of salivary gland stem/progenitor cells. Moreover, we demonstrated a conserved role of Notch signaling in regulating the regenerative capacities of both mouse and human stem/progenitor cells in organoids derived from exocrine and endocrine glandular tissues.

Compared to other, more differentiated SGO models (Yoon et al, 2022), this model shows an enrichment of stem/progenitor cells, providing a valuable platform for the identification of markers of these rare populations, which are often difficult to detect and study in tissue samples. Indeed, scRNA-seq analysis revealed two distinct populations of salivary gland stem/progenitor cells, characterized by increased expression of several markers including *Sox9*, *Foxc1*, and *Cd24a* (stem/progenitor I), and *Itgb1*, *Cd44*, and *Trp63* (stem/progenitor II). These genes have been previously identified as essential regulators of salivary gland development and branching morphogenesis processes (Tanaka et al, 2021, 2022; Wang et al, 2021a; Min et al, 2020). Furthermore, *Sox9*-expressing cells have been shown to be involved in tissue regeneration in several organs following injury (Aggarwal et al, 2024; Xu et al, 2022; Chen et al, 2021), highlighting the regenerative stem cell features of this population. By mapping these genes into a scRNA-seq dataset of freshly isolated salivary gland tissue, we identified a small subpopulation of duct cells enriched with stem/progenitor markers. Although further studies are required to evaluate the presence of these cells in vivo, our results support the notion that a pool of active stem/progenitor cells might reside within the ductal compartment of adult salivary glands (Rocchi et al, 2021a). A

more precise in vivo mapping of these cells could enhance our understanding of the regions to spare during radiotherapy of head and neck cancer, potentially improving both conventional and advanced radiotherapy techniques such as proton therapy (Van Luijk et al, 2015). Moreover, since these organoids are primarily composed of more "immature" epithelial cells, likely due to the high concentration of embryonic laminin in the Matrigel used for their culture (Hughes et al, 2010), developing more physiologically relevant organoid models, such as air liquid interface (ALI) organoids that incorporate stromal and immune components (Santos et al, 2024), could provide deeper insights into the regulation of these cells and their response to irradiation within a more representative tissue context.

*Itgb1* and *Cd44* marked a population of stem/progenitor cells with increased organoid formation and self-renewal capacity. Notably, this population showed elevated expression of acinar-like markers over time, suggesting a potential role as pro-acinar precursor cells. Moreover, this cluster of cells also exhibited markers typical of mesenchymal-like cells, such as *Cd44, Alcam* (CD166), and *Nt5e* (CD73) (García-Bernal et al, 2021), and thus likely represents a plastic population of cells with trans-differentiation and regeneration capacities (Phinney and Prockop, 2007). Although these markers are shared across different cell types of various tissues, the wound healing assay clearly demonstrated the migratory potential of this population. Interestingly, the migratory potential of salivary gland stem cells may also explain the complete loss of homeostasis observed when only the top part of the salivary gland, which contains the tissue stem cells, is irradiated (Van Luijk et al, 2015). Together with their enhanced stemness and differentiation properties, this identifies a novel salivary gland cell population with both stem/progenitor and migratory features, which could potentially improve the efficiency of autologous transplantation and improve salivary gland regeneration following irradiation. Interestingly, although Itgb1/Cd44-expressing cells exhibited clear stem cell properties, RNA velocity and pseudotime analyses identified basal cells as precursor cells for this population. This suggests a potential degree of cellular plasticity and regenerative capability within basal cells. This finding is consistent with previous studies (May et al, 2018; Kwak et al, 2018) demonstrating that salivary gland basal cells can exhibit progenitor-like behavior and contribute to tissue regeneration

following damage. While further studies are needed to validate these observations, this opens the possibility that, similar to basal cells in the airway and mammary glands (Lv et al, 2024; Han et al, 2022), salivary gland basal cells may also retain stem/progenitor characteristics and latent plasticity that can be reactivated under specific conditions such as stress or (radiation) injury.

Notch signaling is a major pathway involved in stem/progenitor cell maintenance and differentiation across several tissues and organs (Siebel and Lendahl, 2017). While its developmental role in guiding the differentiation of multipotent precursors into salivary gland luminal and acinar cells has been established (Chatzeli et al, 2023), our work reveals a critical function for Notch signaling in maintaining adult salivary gland stem/progenitor cell activity under regenerative conditions. Our findings, which extend to various murine and human-derived organoids, align with previous studies (Wang et al, 2021b; Yoon et al, 2022; Ludikhuize et al, 2020; Kessler et al, 2015) demonstrating that the inactivation of the Notch pathway induces terminal differentiation of stem/progenitor cells in different tissues. Furthermore, in line with prior research establishing a critical role of Notch signaling in regulating thyroid and mammary gland function (Mosteiro et al, 2023; Bouras et al, 2008), we demonstrated its significant role in maintaining the self-renewal potential of both TGOs and MGOs. In addition to its roles in self-renewal and differentiation, we uncovered a novel function of Notch signaling in enhancing the regenerative and migration capacities of murine and human salivary gland stem/progenitor cells following irradiation, which could improve future radiotherapy applications. Supporting this notion, recent work has shown that maintaining a balanced Notch signaling is essential for coordinating lung regeneration and preserving mesenchymal/stem cell balance during alveolar repair (Jones et al, 2024). Nevertheless, given the modest upregulation of JAG1 and activated NOTCH1 observed after photon irradiation in mSGOs, it remains important to determine whether additional regulatory mechanisms or cross-talk with other signaling pathways support Notch signaling in maintaining the stem cell niche after injury. Intriguingly, cell–cell interaction analysis uncovered a higher enrichment of Notch signaling after proton irradiation, suggesting enhanced stem/progenitor cell maintenance and a potential advantage of proton therapy over conventional photon-based radiotherapy. Previous studies have shown a link between Notch signaling and radiation-induced DNA damage (Giuranno et al, 2019), which might influence later stem/progenitor cell responses. Notably, in our previous work (Cinat et al, 2025), we demonstrated that Sox9-expressing cells exhibit a higher interferon beta (IFNB) response and increased compensatory proliferation following proton irradiation at later timepoints. Although further studies are required to establish a connection between Notch signaling, IFNB signaling, and stem/progenitor cell activity, we speculate that early activation of Notch signaling could influence radiation-induced immunogenic responses of normal tissue stem/progenitor cells (Fazio and Ricciardiello, 2016), thereby influencing their late response and self-renewal capacities. Additionally, while our work has identified novel putative salivary gland stem/progenitor cells, further studies are required to validate the presence and function of these cell populations in vivo and understand how a more complex microenvironment influences Notch signaling, stem/progenitor cell activity, plasticity, and motility under stress conditions. Furthermore, efforts should be made to identify more specific surface markers for these populations, as CD24 and CD29 expression was detected across a broad range of cells. Lastly, although both male and female patients were included in the study, the use of only female mice may limit the generalizability of the findings; therefore, incorporating male mice in future studies could help avoid potential sex-related biases. However, this consideration is mainly relevant for basic mouse studies, as male mice possess an additional glandular compartment, the glandular convoluted tubules, not present in human salivary glands of female mice.

In summary, our work highlights the complexity of mSGOs and reveals the presence of novel stem/progenitor cell populations within these organoids. Additionally, it underscores the essential role of Notch signaling in regulating the regenerative capacity of murine and patient-derived stem/progenitor cells in different glandular mouse and human organoid models.

# Methods

**Reagents and tools table**

| Reagent/resource | Reference or source | Identifier or catalog number |
|---|---|---|
| **Antibodies** | | |
| FITC anti-rat CD29 (FACS 1:200) | BD biosciences | Cat #555005 |
| PB anti-mouse CD24 (FACS 1:200) | Biolegend | Cat #101820 |
| PE anti-mouse CD44 (FACS 1:200) | BD biosciences | Cat #553134 |
| Rabbit anti-Activated NOTCH1 (WB 1:1000) | Abcam | Cat #ab8925 |
| Rabbit anti-JAG1 (WB 1:1000) | Abcam | Cat #ab300561 |
| Mouse anti-GAPDH (WB 1:10000) | Fitzgerald | Cat #10R-G109a |
| ECL anti-rabbit IgG HRP (WB 1:5000) | GE-Healthcare | Cat #NA934 |
| ECL anti-mouse IgG HRP (WB 1:5000) | GE-Healthcare | Cat #NXA931V |
| Mouse anti-KRT14 (IF 1:300) | Abcam | Cat #ab7800 |
| Rat anti-KRT8/18 (IF 1:50) | DSHB | Cat #ab531826 |
| Rabbit anti-CD29 (IF 1:200) | Cell Signaling | Cat #34971 |
| Rat anti-CD44 (IF 1:200) | Biolegend | Cat #103002 |
| Rabbit anti-AQP5 (IF 1:300) | Abcam | Cat #ab92320 |
| Rabbit anti-SOX9 (IF 1:200) | Cell Signaling | Cat #82630 |
| Mouse anti-CD164 (IF 1:300) | Santa Cruz | Cat #sc-271179 |
| Rabbit anti-NIS (Slc5a5) (IF 1:100) | Proteintech | Cat #24324-1-AP |
| Rabbit anti-Thyroglobulin (TG) (IF 1:100) | Dako | Cat #A0251 |

| Reagent/resource | Reference or source | Identifier or catalog number |
|---|---|---|
| **Oligonucleotides and other sequence-based reagents** | | |
| rt-qPCR primers | This study | Appendix Table S1 |
| **Chemicals, enzymes and other reagents** | | |
| DMEM/F12 | Gibco/Invitrogen | Cat #11320-074 |
| Penicillin–streptomycin antibiotics | Invitrogen | Cat #15140-163 |
| Glutamax | ThermoFisher Scientific | Cat #35050038 |
| EGF | Sigma-Aldrich | Cat #E9644 |
| FGF2 | Peprotech | Cat #100-18-B |
| N2 | Gibco | Cat #17502-048 |
| Insulin | Sigma-Aldrich | Cat #I6634-100MG |
| Dexamethasone | Sigma-Aldrich | Cat #d4902-25mg |
| 0.05% trypsin EDTA | Invitrogen | Cat #25300-096 |
| Matrigel | Vwr | Cat #356235 |
| Y27632 | Abcam | Cat #ab120129 |
| Noggin | Preprotech | Cat #120-10 C |
| A8301 | Tocris Bioscience | Cat #2939 |
| Collagenase type II | Gibco | Cat #17101-015 |
| Hyaluronidase | Sigma | Cat #H3506-5G |
| $CaCl_2$ | Sigma | Cat #C3306 SIGMA |
| HBSS | Gibco | Cat #14175-129 |
| BSA 7.5% | Gibco | Cat #15260037 |
| Fetal bovine serum | Bodinco | Cat #500 ml |
| HGF | Preprotech | Cat #100-39 |
| B27 supplement | Gibco | Cat #17504001 |
| CTS B27 supplement XenoFree | Gibco | Cat #A5047501 |
| R-Spondin-1 human | Sigma | Cat #SRP3292-20UG |
| UltraGRO-PURE GI Cell Culture Supplement Xeno-free | Pelobiotech | Cat #PB-HPCHXCGLI05 |
| HEPES | Thermo Scientific | Cat #15630080 |
| ITS | Gibco | Cat #41400045 |
| bTSH | Sigma | Cat #T8931 |
| FGF10 | PeproTech | Cat #100-26 |
| Advanced DMEM/F12 | Gibco/Invitrogen | Cat #12634028 |
| Nicotinamide | Sigma | Cat #N0636 |
| N-acetylcysteine | Sigma-Aldrich | Cat #A9165-5G |

| Reagent/resource | Reference or source | Identifier or catalog number |
|---|---|---|
| Primocin | InvivoGen | Cat #ant-pm-05 |
| Hydrocortisone | Cayman Chemical | Cat #20739 |
| β-estradiol | Sigma-Aldrich | Cat #E2257 |
| Forskolin | BioGems | Cat #6652995 |
| Heregulin β1 | PeproTech | Cat #100-03 |
| SB202190 | Cayman Chemicals | Cat #10010399-25 |
| FGF7 | Immunotools | Cat #11343653 |
| Culturex growth factor reduced BME | R&D Systems | Cat #3433-010-01 |
| DBZ | Axon Medchem | Cat #1488 |
| JAG1 | Alpha Diagnostic | Cat #SP-56614-1 |
| DMSO | Sigma | Cat #276855-100 ML |
| DAPT | Sigma | Cat #D5942-5MG |
| Propidium iodide | Sigma | Cat #P4170 |
| TGX Stain-Free™ FastCast™ Acrylamide Kit, 12% | Biorad | Cat #1610185 |
| HistoGel embedding medium | Epredia | Cat #HG-4000-012 |
| Tris base | Merck | Cat #10708976001 |
| EDTA | Invitrogen | Cat #15576-028 |
| Donkey serum | Jackson Immuno Research | Cat #017-000-121 |
| Triton X-100 | Sigma | Cat #T8787-250ML |
| DAPI | Sigma-Aldrich | Cat #D9542 |
| Mounting medium | Dako | Cat #S3025/149699 |
| dNTP mix | Invitrogen | Cat #10297-018 |
| Random primers | Invitrogen | Cat #SO142 |
| First-strand Buffer | Invitrogen | Cat #28025013 |
| DTT | Invitrogen | Cat #328025013 |
| RNase OUT™ | Invitrogen | Cat #10777019 |
| M-MLV RT | Invitrogen | Cat #28025013 |
| iQ SYBR Green Supermix | Bio-Rad | Cat #170-8885 |
| **Software** | | |
| FlowJo (v10.10.0) | https://flowjo.com/flowjo/overview | |
| Image Lab (v6.1) | https://www.bio-rad.com/en-us/product/image-lab-software?ID=KRE6P5E8Z | |

| Reagent/resource | Reference or source | Identifier or catalog number |
|---|---|---|
| ImageJ (v1.52) | https://imagej.en.softonic.com/download | |
| GraphPad Prism (v8.0.1) | https://www.graphpad.com/featuresw | |
| **Other** | | |
| 10X Chromium Next GEM Single Cell 3' Kit v3.1 | 10X Genomics | Cat #1000128 |
| Single Index Kit T Set A | 10X Genomics | Cat #1000213 |
| 10X Chromium Next GEM Chip G Single Cell Kit | 10X Genomics | Cat #1000127 |
| Nextera DNA Sample Preparation Kit | Illumina | Cat #FC-121-1030 |
| Qiagen MinElute kit | Qiagen | Cat #28004 |
| E-gel agarose gel | ThermoFisher Scientific | Cat #G521802 |
| Zymoclean Gel DNA Recovery Kit | Zymo | Cat #D4007 |
| NextSeq 500 | Illumina | |
| Leica DM6 microscope | Leica Microsystems | |
| Leica DMI 8 Inverted microscope | Leica Microsystems | |
| ChemiDoc imaging system | Bio-Rad | |
| Moflo Astrios Cell Sorter | Beckman Coulter | |
| Cytek Aurora CS | Cytek Biosciences | |
| Bio-Rad Real-Time PCR System | Bio-Rad | |
| Qubit | ThermoFisher Scientific | |
| Bioanalyzer | Agilent | |

## Mice

Given the known anatomical differences between male and female mice, particularly the presence of granular convoluted tubules within the submandibular glands of males, which are absent in humans, only female mice were used for this study. Submandibular salivary glands and thyroid glands were collected from 8- to 12-week-old female C57BL/6 mice (Envigo, the Netherlands). Mice were housed under standard conditions with diet and water ad libitum. Animal work was approved by the Central Committee of Animal Experimentation of the Dutch government and the Institute Animal Welfare Body of the University Medical Center Groningen (UMCG) [animal welfare body (IVD) protocol number 184824-01-001].

## Patient biopsies

Human non-malignant submandibular gland, thyroid gland and mammary tissues were obtained from donors after informed consent and Institutional Review Board (IRB) approval during scheduled surgery for the removal of squamous cell carcinoma of the oral cavity (for the submandibular gland), papillary thyroid carcinoma (for the thyroid gland) and invasive ductal carcinoma (for the mammary gland) at the University Medical Center Groningen (UMCG), Medical Centre Leeuwarden and Martini Ziekenhuis.

## Organoid cultures

*Mouse salivary gland organoids:* mouse submandibular salivary glands were first mechanically and enzymatically digested in digestion buffer [collagenase type II (0.63 mg/mL), hyaluronidase (0.5 mg/mL) and $CaCl_2$ (6.25 mM) in HBSS/BSA 1%] and then cultured in minimal medium (MM) consisting of DMEM/F12, penicillin–streptomycin (Pen/Strep) antibiotics, glutamax (2 mM), EGF (20 ng/ml), FGF2 (20 ng/ml), N2 (1×), insulin (10 μg/ml) and dexamethasone (1 μM). After 3 days in culture, mouse salivary spheres were dissociated into single cells using 0.05% trypsin EDTA and counted. In all, 10,000 mouse cells were then plated in 75 μL gel/well [25 μL cell suspension + 50 μL of Matrigel] in a 12-well tissue culture plate. After solidification of the gels, 1 mL of WRY medium [MM, Y27632 (10 μM), 10% R-spondin1-conditioned medium, and 50% Wnt3a-conditioned medium] was added to each well. The plates were then placed at 37 °C and 5% $CO_2$ until the day of the experiment.

*Human salivary gland organoids:* human submandibular salivary gland biopsies were mechanically and enzymatically digested in digestion buffer and counted. In all, 20,000 cells were then plated in 75 μL gel/well [25 μL cell suspension + 50 μL of Matrigel] in a 12-well tissue culture plate. After solidification of the gels, 1 mL of WRYTN medium [MM medium, Y27632, Noggin (50 ng/mL), A8301 (1 μM), 10% R-spondin1-conditioned medium, and 50% Wnt3a-conditioned medium] was added to each well of human cells. The plates were then placed at 37 °C and 5% $CO_2$ until the day of the experiment.

*Mouse thyroid gland organoids:* as previously described (Ogundipe et al, 2021), following mechanical and enzymatic digestion mouse samples were cultured in thyroid gland medium (TGM) consisting of DMEM/F12, Pen/Strep antibiotics, glutamax, EGF, FGF2, and 0.5% B27 supplement. One day after isolation, mouse thyroid spheres were dissociated into single cells using 0.05% trypsin EDTA and counted. In all, 10,000 mouse cells were then placed in 75 μL gel/well [25 μL cell suspension + 50 μL of Matrigel] in a 12-well tissue culture plate and 1 mL of TGM supplemented with Y27632 was added to each well. The plates were placed at 37 °C and 5% $CO_2$ until the day of the experiment.

*Human thyroid gland organoids:* as previously described (Ogundipe et al, 2021), following mechanical and enzymatic digestion, human thyroid gland cells were counted and 25,000 cells were placed in 75 μL gel/well [25 μL cell suspension + 50 μL of Matrigel] in a 12-well tissue culture plate. 1 mL of human TGM (hTGM) consisting of DMEM/F12, Pen/Strep antibiotics, glutamax, R-Spondin-1 human (100 ng/mL), 10% UltraGRO-PURE GI Cell Culture Supplement Xeno-free, EGF (40 ng/mL), FGF2, 0.5% CTS B27 supplement XenoFree, Y27632, A8301 (5 μM), nicotinamide (10 mM) and 1% Heparin Sodium Salt Solution was then added to each well. The plates were then placed at 37 °C and 5% $CO_2$ until the day of the experiment.

*Mammary gland organoids:* mammary gland biopsies were first mechanically and enzymatically digested in digestion buffer

consisting of mammary gland medium (MGM) [Advanced DMEM, Pen/Strep antibiotics, glutamax, 20% Wnt3a-conditioned medium, 10% R-spondin conditioned medium, 1× B27 supplement, nicotinamide, Noggin (25 ng/mL), Y27632 (5 μM), N-acetylcysteine (1.25 mM), Primocin (100 μg/mL), Hydrocortisone (500 ng/mL), β-estradiol (100 nM), Forskolin (10 μM), Heregulin β1 (5 nM), EGF (5 ng/mL), SB202190 (1 μM), FGF7 (5 ng/mL), FGF10 (20 ng/mL), A8301 (0.5 μM) and 1% Heparin Sodium Salt Solution] and Collagenase II (1 mg/mL). Human mammary gland cells were then counted and 10,000 cells were placed in 80 μL gel/well [20 μL cell suspension + 60 μL growth factor reduced BME extract] in a 12-well tissue culture plate. After solidification of the gels 1 mL of MGM was added to each well. The plates were placed at 37 °C and 5% $CO_2$ until the day of the experiment.

## Self-renewal assay

To assess the self-renewal capacity of the stem/progenitor cells within the mouse and human organoid models, organoids were collected and dissociated into single cells at the proper time point using 0.05% trypsin-EDTA. As previously described, single cells were then placed in Matrigel (SGOs and TGOs) or BME (MGOs) in a 12-well tissue culture plate with 1 mL of full medium. Plates were placed at 37 °C and 5% $CO_2$. Between 7 and 14 days later (based on the organoid model), organoids were collected and dissociated into single cells. Organoid and single cell numbers were noted and used to calculate the organoid formation efficiency (OFE) and population doubling (PD) as follows:

$$OFE\% = (\text{total number of organoids harvested/number of seeded cells}) \times 100$$

$$PD = \ln(\text{total number of harvested cells/number of seeded cells})/\ln2$$

## Organoid differentiation

To assess the differentiation capacity of both mouse and thyroid gland organoids, 7-day organoids were carefully harvested and counted. Approximately 30–50 organoids were then placed in a 75 μL gel/well [25 μL cell suspension + 50 μL of Matrigel] in a 96-well tissue culture plate previously coated with a layer of Matrigel diluted 1:1 (20 μL DMEM/F12 + 20 μL Matrigel). After solidification of the gels, 150 μL of salivary gland differentiation media [MM, 10% FBS, HGF (50 ng/mL), ±DBZ (1 μM)] or 150 μL of thyroid gland differentiation media [TGM, 10% FBS, Heparin Sodium Salt Solution (100 ng/mL), IGF1 (50 ng/ml), FGF10 (100 ng/mL), insulin (5 μg/ml), ITS (5 μg/ml), Dexamethasone (50 nM), bTSH (100 mU/mL), ±DBZ (1 μM)] were added to each well and refreshed every 2–3 days. The plates were then placed at 37 °C and 5% $CO_2$. After 8 days in culture, the organoids were collected and processed for rt-qPCR or fixed and embedded in paraffin for immunofluorescence staining.

## Treatments

Five-day salivary gland organoids were irradiated either with 7 Gy photons or protons. Photon irradiation was performed using a Cesium-137 source with a dose rate of 0.59 Gy/min at the Department of Biomedical Sciences of the UMCG. Proton irradiation was performed with a 150 MeV proton beam at the Particle Therapy Research Center (PARTREC) accelerator facility of the UMCG or at the UMCG Proton Therapy Center (GPTC). Samples were placed in the plateau region of a 150 MeV Bragg curve.

To assess the role of Notch signaling in irradiated samples Dibenzoazepine (DBZ), JAG1 or equivalent volume of solvent (DMSO) were added to salivary gland organoids at a final concentration of 1 μM. To evaluate the role of Notch signaling in control condition, organoids from all glands were cultured without Wnt3a-conditioned medium [Enriched Medium (EM)]. 1 μM of DBZ was added to the medium from day 0 and refreshed every 4 days until the end of the experiment. mSGOs were also treated with 1 μM of DAPT.

## Fluorescence activated cell-sorting

For the sorting of stem/progenitor cells, mouse salivary and thyroid gland organoids were harvested and dissociated into single cells using 0.05% trypsin EDTA. Cells were resuspended in 0.2% PBS/BSA and incubated with the proper conjugated primary antibody for 30 min at 4 °C. After washing, cells were resuspended in 0.2% PBS/BSA containing propidium iodide (PI; 1 mg/mL) and sorted with the Moflo Astrios sorter or the Cytek Aurora CS at the Flow Cytometry Unit (FCU) at the UMCG. Sorted cells were washed and placed in a 75 μL gel/well [25 μL cell suspension + 50 μL of Matrigel] at a concentration of 10,000 cells/gel in a 12-well tissue culture plate. 1 mL of WRY medium (SGOs) or TGM medium (TGOs) was added to each well and the plates were then placed at 37 °C and 5% $CO_2$. Organoids were harvested and counted at day 7 for OFE calculation. Flow cytometry analysis was performed using the FlowJo software (v10.10.0).

## Western blot

Following harvesting, mSGOs were lysed in RIPA buffer, sonicated for 10 s, and centrifuged at maximum speed for 5 min at 4 °C. Protein concentrations in the resulting lysates were determined using the Bradford assay. The samples were then boiled at 99 °C for 5 min prior to gel loading. Equal amounts of protein were resolved on 10% polyacrylamide gels and transferred onto nitrocellulose membranes using the Trans-Blot Turbo System (Bio-Rad). Membranes were blocked with 10% milk in PBS-Tween20 for 30 min. After sectioning, membranes were incubated with primary antibodies overnight at 4 °C, followed by a 1.5-h incubation at room temperature with horseradish peroxidase-conjugated secondary antibodies. Detection was performed using ECL reagent on a ChemiDoc imaging system (Bio-Rad), and signal quantification was carried out using Image Lab software.

## Wound healing assay

Control and irradiated salivary gland and thyroid gland organoids were collected and dissociated into single cells using 0.05% trypsin-EDTA. After cell counting, 90,000 cells/well were re-seeded in a 24-well plate pre-coated with Matrigel. WRY medium (for salivary gland cells) or TGM medium (for thyroid gland cells) was added, and cells were incubated at 37 °C with 5% $CO_2$. Once cells reached confluency, they were starved by replacing the full medium with DMEM/F12

without growth factors for at least 3 h. A wound was then created in each well, and fresh DMEM/F12 containing either DBZ (1 µM), JAG1 (1 µM), or DMSO was added. Plates were then placed in an IncuCyte S3 Live-Cell Analysis System for imaging. After 24 h, cells were fixed with 4% formaldehyde at room temperature for 10 min and then permeabilized using 0.2% Triton for 4 min. Primary antibody diluted in blocking solution (2% BSA) was then added to each well and incubated at room temperature for 1 h. After washing, appropriate secondary antibody diluted in blocking solution (2% BSA) was added to each well at room temperature for 1 h and with DAPI at room temperature for 10 min for nuclear staining. Images were then acquired using a Leica DMI 8 Inverted microscope. Images were processed and analyzed using ImageJ (v1.52).

## Immunofluorescence staining

For the staining of organoid sections, salivary and thyroid gland organoids were collected, fixed with 4% formaldehyde for 15 min and transferred into a drop of HistoGel embedding medium. HistoGels were then dehydrated, embedded in paraffin and cut into 4 µm thick sections. For the staining, sections were dewaxed and cooked with Tris-EDTA (pH 9) antigen retrieval buffer. After permeabilization and blocking [4% donkey serum, 1% BSA, 0.01% Triton in PBS 1×], sections were incubated with the proper primary antibody at 4 °C overnight. On the next day, sections were incubated with the secondary antibody at room temperature for 1 h and with DAPI at room temperature for 10 min for nuclear staining. Upon mounting, images were acquired with a Leica DM6 microscope. Images were processed using ImageJ (v1.52).

## Quantitative real-time qPCR

Isolation of total RNA from salivary and thyroid gland organoids was performed using the RNeasy Mini Kit according to the manufacturer's protocol. RNA was retro transcribed by using 1 µL dNTP mix (10 mM), 1 µL random primers (100 ng), 4 µL 5× First-strand Buffer, 2 µL DTT (0.1 M), 1 µL RNase OUTTM (40 units/µL), and 1 µL M-MLV RT (200 units). To assess the expression of the genes of interest, specific primers were used together with iQ SYBR Green Supermix. All reactions were run in triplicate on a Bio-Rad Real-Time PCR System. The list of primers is specified in Appendix Table S1. The *Ywhaz* gene was used as internal control.

## Single-cell RNA sequencing library preparation

The preparation of the single-cell RNA sequencing library was performed as previously described (Cinat et al, 2025). In brief, upon harvesting and dissociation of the salivary gland organoids, 45,000 propidium iodide negative cells/sample were sorted using the Moflo Astrios sorter at the FCU at the UMCG. Samples were then loaded on the Chromium Next GEM Chip G and ran in the Chromium Controller (10X Genomics). Library construction was performed following the 10X Genomics protocol and library quality control and concentration measurement were performed using the Qubit and Tapestation machines at the Research Sequencing Facility of ERIBA at the UMCG. For the sequencing, libraries were equimolarly pooled and 1.8 pM of the pool with 5% PhiX were loaded on a NextSeq 500 (Illumina) for a 75 bp paired-end sequencing run at the Research Sequencing Facility of ERIBA (UMCG).

## Single-cell RNA sequencing analysis

Reads without cell barcodes or UMIs were excluded and remaining raw reads were aligned to the mouse genome GRCm39. Following demultiplexing and cell filtering, the remaining cells were used for downstream analysis. Single-cell RNA sequencing analysis was done on R (v4.3.2) using the Seurat package (v5.0.3) (Hao et al, 2024). Cells were excluded based on their mitochondria counts (more than 5%) and feature counts (less than 2500). The principal components (PC) to use for UMAP dimensionality reduction for each sample were calculated using the Elbow plots and Jack Straw. Doublets and clusters with less than 50 cells with no significant markers were excluded from the analysis and cluster-markers were identified using the FindAllMarkers function with the statistical test MAST. The CellChat package (v2.1.2) (Jin et al, 2021) was used for cell–cell interaction analysis with standard settings. Harmony (Korsunsky et al, 2019) (v.1.2.0) was used for the integration of 7-day and 11-day mSGOs after regressing out cell cycle effects. Velocity and pseudotime analyses were performed on Python (v3.8.10) using the packages velocyto (v0.17.16) (La Manno et al, 2018), scvelo (v0.3.3) (Bergen et al, 2020) and monocle3 (v1.4.25) (Trapnell et al, 2014).

## Bulk RNA-sequencing analysis

Analysis of bulk RNA-seq data from (Cinat et al, 2025) was performed as previously described (Cinat et al, 2025). In brief, for the alignment of the fastq files the mouse genome GRCm39 was used. R (v4.3.2) and RStudio were used for the downstream analysis. For the differential expression analysis, low expressed genes (total count <1 in less than 2 samples) were excluded. The R package edgeR (Robinson et al, 2009) was used for normalization and identification of differential expressed genes (LogFC > 0.6, FDR < 0.05). CIBERSORTx was used for deconvolution (Newman et al, 2019).

## ATAC-sequencing library preparation and analysis

ATAC-sequencing (ATAC-seq) library preparation was performed as previously described (Kracht et al, 2020). In brief, mSGOs derived from four different mice were harvested. ATAC-seq libraries were then prepared using Nextera DNA Sample Preparation Kit (Illumina, FC-121-1030) following the manufacturer's protocol. Following PCR amplification, DNA was purified using the Qiagen MinElute kit and run on a 2% E-gel agarose gel (ThermoFisher Scientific, G521802). The concentration of the libraries was determined using a Qubit (Thermo-Fisher Scientific) and 2100 Bioanalyzer (Agilent). In all, 1.6 pM of libraries with 5% PhiX was then loaded on a NextSeq 500 (Illumina) for a 75 bp paired-end sequencing run at the Research Sequencing Facility of ERIBA (UMCG).

For the analysis, alignment and peak calling were performed using the ENCODE ATAC-seq pipeline. All organoid samples were used to generate a consensus peak file (>41 K peaks). HOMER (Li et al, 2009) function findMotifsGenome was used to perform motif enrichment analysis with default settings. Transcription factors associated to each motif were integrated with the single-cell RNA sequencing dataset and their average expression within each cell cluster was calculated.

## Graphics

Synopsis and schematic graphics in main and appendix figures were created with BioRender.com.

## Statistical analysis

The number of biological replicates and the details of the statistical tests used in each experiment are stated in main and appendix figure legends. Data is shown as mean ± s.e.m., the Shapiro–Wilk test was used to test normality of distribution of raw data. Two-sided unpaired and paired *t* test was used for two group comparisons while one-way ANOVA and two-way ANOVA with appropriate post-hoc test were used for multiple (>2) group comparisons. Sample sizes were estimated empirically and *p* values ≤ 0.05 were considered statistically significant. No blinding was performed. Statistical analyses were performed on raw data using the GraphPad Prism software (v8.0.1).

## Data availability

The sequencing data have been deposited in the Gene Expression Omnibus (GEO) repository under accession numbers GSE272759 (scRNA-seq), GSE273394 (ATAC-seq) and GSE303285 (bulk RNA-seq). All other data are stored at the Department of Biomedical Sciences, UMCG, and are available from the corresponding authors upon request. GSE272759; GSE273394; GSE303285.

The source data of this paper are collected in the following database record: biostudies:S-SCDT-10_1038-S44318-025-00607-w.

## Peer review information

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

## Acknowledgements

We would like to thank the surgical teams of the University Medical Center Groningen, Medical Center Leeuwarden and Martini Ziekenhuis for providing donor biopsies. We also thank S Both, J Free and BN Jones for technical support at the PARTREC and GPTC; B Eggen, N Brouwer and E Gerrits for support in the scRNA-seq library preparation and analysis; the members of the ERIBA Research Sequencing Facility (UMCG) for the sequencing runs; T Bijma and J Teunis from the Flow Cytometry Unit (UMCG) for the sorting. Illustrations were created with Biorender.com. This work was supported by the Dutch Cancer Society KWF Grant nr 12092 and IBA Grant nr PPP-2021-27.

## Author contributions

**Davide Cinat**: Conceptualization; Data curation; Formal analysis; Investigation; Visualization; Methodology; Writing—original draft; Writing—review and editing. **Rufina Maturi**: Formal analysis. **Jeremy P Gunawan**: Formal analysis. **Anne L Jellema-de Bruin**: Formal analysis. **Laura Kracht**: Formal analysis. **Paola Serrano Martinez**: Formal analysis. **Yi Wu**: Formal analysis. **Abel Soto-Gamez**: Resources; Formal analysis; Writing—original draft. **Marc-Jan van Goethem**: Methodology. **Inge R Holtman**: Resources; Formal analysis. **Sarah Pringle**: Resources; Writing—original draft. **Lara Barazzuol**: Conceptualization; Resources; Data curation; Supervision; Visualization; Methodology; Writing—original draft; Project administration; Writing—review and editing. **Rob P Coppes**: Conceptualization; Resources; Data curation; Supervision; Funding acquisition; Visualization; Methodology; Writing—original draft; Project administration; Writing—review and editing.

Source data underlying figure panels in this paper may have individual authorship assigned. Where available, figure panel/source data authorship is listed in the following database record: biostudies:S-SCDT-10_1038-S44318-025-00607-w.

## Disclosure and competing interests statement

The authors declare no competing interests.

