## [Peer Review File · The EMBO Journal]

Notch signaling is a driver of glandular stem cell activity and regenerative migration after damage

Davide Cinat, Rufina Maturi, Jeremy Gunawan, Anne Jellema - de Bruin, Laura Kracht, Paola Serrano Martinez, Yi Wu, Abel Soto-Gamez, Marc-Jan van Goethem, Inge Holtman, Sarah Pringle, Lara Barazzuol, and Rob Coppes

Corresponding authors: Rob Coppes (r.p.coppes@umcg.nl), Lara Barazzuol (l.barazzuol@umcg.nl)

Review Timeline:

Submission Date:	5th Feb 25
Editorial Decision:	17th Mar 25
Revision Received:	30th Jun 25
Editorial Decision:	12th Sep 25
Revision Received:	22nd Sep 25
Accepted:	6th Oct 25

Editor: Daniel Klimmeck

Transaction Report:

Dear Dr Coppes,

Thank you again for the submission of your manuscript (EMBOJ-2025-120401) to The EMBO Journal. Please accept my apologies for the unusual delay with the peer-review of your manuscript at this time of the year. Your manuscript was sent to two reviewers, and we have now received reports from the both of them, which I enclose below. Based on all information at hand, we have now decided to invite you to revise your work for the EMBO Journal.

As you will see from the experts' reports, the referees acknowledge the analysis and potential interest and value of your findings. However, they also express important issues, which need to be addressed thoroughly to make them supportive of publication in the EMBO Journal. Further, the reviewers raise a number of issues related to the presentation of the findings, statistics applied and overall discussion of related literature, that would need to be conclusively addressed to achieve the level of robustness and clarity needed for The EMBO Journal.

Given the overall interest stated and broader angle of your findings, we are able to invite you to revise your manuscript experimentally to address the referees' comments. I need to stress though that we do require strong support from the referees on a revised version of the study in order to move on to publication of the work.

I would appreciate if you could contact me during the next weeks for exchange e.g. a video call to discuss your perspective on the comments and potential plan for revisions.

Please feel free to contact me if you have any questions or need further input on the referee comments.

When submitting your revised manuscript, please carefully review the instructions below.

Please feel free to approach me any time should you have additional questions related to this.

Thank you for the opportunity to consider your work for publication.

I look forward to your revision.

Kind regards,

Daniel Klimmeck

Daniel Klimmeck, PhD
Senior Editor
The EMBO Journal

Instruction for the preparation of your revised manuscript:

- 1) a .docx formatted version of the manuscript text (including legends for main figures, EV figures and tables). Please make sure that the changes are highlighted to be clearly visible.
- 2) individual production quality figure files as .eps, .tif, .jpg (one file per figure).
- 3) a .docx formatted letter INCLUDING the reviewers' reports and your detailed point-by-point response to their comments. As part of the EMBO Press transparent editorial process, the point-by-point response is part of the Review Process File (RPF), which will be published alongside your paper.
- 4) a complete author checklist, which you can download from our author guidelines ([https://wol-prod-cdn.literatumonline.com/pb-assets/embo-site/Author Checklist%20-%20EMBO%20J-1561436015657.xlsx](https://wol-prod-cdn.literatumonline.com/pb-assets/embo-site/Author%20Checklist%20-%20EMBO%20J-1561436015657.xlsx)). Please insert information in the checklist that is also reflected in the manuscript. The completed author checklist will also be part of the RPF.

6) It is mandatory to include a 'Data Availability' section after the Materials and Methods. Before submitting your revision, primary datasets produced in this study need to be deposited in an appropriate public database, and the accession numbers and database listed under 'Data Availability'. Please remember to provide a reviewer password if the datasets are not yet public (see <https://www.embopress.org/page/journal/14602075/authorguide#datadeposition>).

7) Our journal encourages inclusion of *data citations in the reference list* to directly cite datasets that were re-used and obtained from public databases. Data citations in the article text are distinct from normal bibliographical citations and should directly link to the database records from which the data can be accessed. In the main text, data citations are formatted as follows: "Data ref: Smith et al, 2001" or "Data ref: NCBI Sequence Read Archive PRJNA342805, 2017". In the Reference list, data citations must be labeled with "[DATASET]". A data reference must provide the database name, accession number/identifiers and a resolvable link to the landing page from which the data can be accessed at the end of the reference. Further instructions are available at .

8) At EMBO Press we ask authors to provide source data for the main and EV figures. Our source data coordinator will contact you to discuss which figure panels we would need source data for and will also provide you with helpful tips on how to upload and organize the files.

Numerical data can be provided as individual .xls or .csv files (including a tab describing the data). For 'blots' or microscopy, uncropped images should be submitted (using a zip archive or a single pdf per main figure if multiple images need to be supplied for one panel). Additional information on source data and instruction on how to label the files are available at .

9) We replaced Supplementary Information with Expanded View (EV) Figures and Tables that are collapsible/expandable online (see examples in <https://www.embopress.org/doi/10.15252/embj.201695874>). A maximum of 5 EV Figures can be typeset. EV Figures should be cited as 'Figure EV1, Figure EV2" etc. in the text and their respective legends should be included in the main text after the legends of regular figures.

11) For data quantification: please specify the name of the statistical test used to generate error bars and P values, the number (n) of independent experiments (specify technical or biological replicates) underlying each data point and the test used to calculate p-values in each figure legend. The figure legends should contain a basic description of n, P and the test applied. Graphs must include a description of the bars and the error bars (s.d., s.e.m.).

We realize that it is difficult to revise to a specific deadline. In the interest of protecting the conceptual advance provided by the work, we recommend a revision within 3 months (15th Jun 2025). Please discuss the revision progress ahead of this time with the editor if you require more time to complete the revisions.

Referee #1:

In this study, Cinat and colleagues utilize single-cell RNA sequencing to profile salivary gland organoids, identifying key cellular populations and signaling pathways. Their findings highlight Sox9- and Itgb1/Cd44-expressing cells as crucial stem/progenitor populations, with Notch signaling playing a central role in self-renewal and migration following irradiation. These results extend to other glandular tissues, emphasizing the conserved function of Notch signaling in stem/progenitor cell-mediated regeneration. Understanding the mechanisms underlying salivary gland regeneration is essential for improving the quality of life of cancer patients and survivors undergoing radiotherapy. While this study is both novel and timely, several conceptual and experimental aspects require further clarification.

Figure 1A: The authors mention that organoids are passaged; however, this is not depicted in the corresponding cartoon.

Figure 1D-H: Specific clusters are identified using markers. However, employing a set of lineage-specific signatures combined with the AddModuleScore function from Seurat could provide a more unbiased approach. Additionally, to assess how closely salivary gland organoids resemble actual tissue, integrating different datasets could help determine how well cellular heterogeneity is recapitulated.

Figure 1F-G: In the scRNA-seq analysis, expression is relative to the other detected clusters. Showing a bubble plot from the integrated data would help support the claims about more mature and primitive states.

Figure 1H: Is this panel showing a reclustering or simply a zoomed-in view? If the goal is to increase resolution, a reclustering approach would be more appropriate.

Figure 1N: The specificity of CD164 expression should be clarified, as it appears ubiquitously expressed in the provided image, which contrasts with the scRNA-seq data.

Figure 2A: The motifs appear specifically enriched at day 7 compared to day 11 in the ATAC-seq data. Including a bubble plot for these genes at day 11 would help contextualize these findings.

Figure 2B: The presence of a day-dependent cluster suggests a potential batch effect. Improving data integration could address this issue.

Figure 2C-D: The authors perform pseudotime analysis, which is conceptually complex given the two distinct time points. One would expect stem/progenitor cells to give rise to differentiated cells at each time point. Improved data integration could help resolve this issue. Another limitation is the method used to define the root of the pseudotime trajectory. Employing more unbiased approaches, such as RNA velocity and CellRank, could improve the identification of initial and terminal states.

Figure 2E: The bulk RNA-seq analysis of salivary gland organoids at days 5, 7, and 11 shows specific lineage and proliferation markers. The authors suggest an enrichment of stem/progenitor cells at day 7; however, the data indicate that days 7 and 11 are quite similar, whereas day 5 appears distinct. Statistical analyses should be applied to confirm whether differences between days 7 and 11 are significant.

Figures 1F-I: The authors state that CD29 and CD24 are highly expressed in the stem/progenitor cluster. However, they are also prominently expressed in the basal and cycling clusters, making it challenging to attribute higher organoid formation efficiency solely to this cluster.

Figure 2M: The authors assess the migratory potential of 2D salivary gland organoids, but the analysis appears qualitative, as different populations are not compared. Would this differ in organoids at day 11? Additionally, CD44+ cells are reported at the migratory front, but are these cells also proliferative?

Figure 3A: After characterizing cell clusters in salivary gland organoids, the authors transition to modeling regeneration via photon and proton irradiation. The experimental design is somewhat unexpected, as irradiation is performed at day 5, a time point that was not previously characterized. Given the prior focus on day 7, it would be more logical to irradiate at day 7 and

analyze responses at day 11. Performing scRNA-seq at day 5 could help resolve this discrepancy.

Figure 3B: The authors claim that photon and proton irradiation do not induce significant differences and analyze the data collectively. However, the UMAP suggests otherwise. Would it be possible to show the proportions of cells in each cluster under different conditions (e.g., using scCODA)?

Figure 3D: The authors perform GSEA to identify enriched pathways in irradiated samples. While GSEA is valuable for generating hypotheses, experimental validation of some observed correlations would strengthen the conclusions.

Figure 3F: The authors present the relative contribution of ligand-receptor interactions in the control condition. Showing how these interactions change upon irradiation would enhance the panel's message.

Figures 3I-J: The authors conclude that Notch-related genes (Jag1, Jag2, Notch1) are upregulated upon irradiation. Are these differences statistically significant? Is there supporting protein-level evidence? Additionally, in the 3D organoid context, where are the sending and receiving cells located? Are there detectable differences in cleaved Notch protein levels or distribution?

Figure 4: To investigate Notch signaling in salivary gland organoid regeneration, the authors inhibit its activation using DBZ. However, they do not provide direct evidence that their treatment effectively inhibits the pathway. Additionally, while functional assays are performed, the transcriptional and cell identity changes induced by Notch inhibition remain unclear, complicating result interpretation.

Figure 4E: The organoids treated with and without DBZ appear different from day 0, which is unexpected. Including whole-well images and quantifications of circularity or budding structures would provide a clearer picture.

Figure 4L: There appears to be a discrepancy between Figures 4L and 4C. Can the authors clarify this? Additionally, in the DMSO-treated condition, the control values are consistently set to 1, suggesting no variability. How were statistical analyses applied in this context? This data processing approach risks inflating small fold-change differences into statistically significant results.

Figures 5K, N, P: Similar to the previous comment, it is unexpected that all controls are normalized to a fold-change of 1. How were statistical analyses performed under this assumption? Showing raw data instead of fold changes might be more informative and would allow conclusions to be adapted accordingly.

Referee #2:

This study provides valuable insights into the cellular composition and regenerative potential of mouse submandibular salivary gland organoids (mSGOs) through single-cell RNA sequencing (scRNA-seq) and functional assays. The analysis identifies distinct cell populations in mSGOs at 7 and 11 days of culture, including two stem/progenitor cell populations, basal and luminal duct cells, and differentiated cell types. The 7-day organoids exhibit a more stem/progenitor-like phenotype, while the 11-day organoids show increased differentiation. Functional assays confirm that stem/progenitor cells from 7-day mSGOs have self-renewal capabilities. Additionally, irradiation studies show that stem/progenitor cells surviving radiation exhibit increased regenerative activity and Notch signaling, highlighting their role in tissue repair. The study addresses important questions regarding the plasticity and resilience of salivary gland cells. However, limitations such as the lack of in vivo and in vitro integration, the absence of cell cycle normalization, and potential sex bias should be considered.

Major Concerns:

- Integration of in vivo and in vitro datasets: To further strengthen the comparison between in vitro and in vivo conditions, it would be valuable to integrate the datasets and perform a direct comparative analysis between the mSGO-derived populations and their corresponding counterparts in native salivary gland tissue. While the study highlights similarities between mSGOs and in vivo tissue, a systematic assessment of transcriptional differences between these conditions could provide deeper insights into how well the organoid model recapitulates the native environment. For instance, identifying differentially expressed genes between mSGO-derived stem/progenitor cells and their in vivo counterparts could reveal culture-induced adaptations or limitations of the in vitro system.
- Cell cycle normalization: The presence of a cycling population suggests that normalization should be performed to ensure that clustering and gene expression differences are not primarily driven by proliferative status.
- Sex bias in study design: Only female mice were used. The authors should provide a rationale for this choice and discuss potential limitations in terms of generalizability.

Minor Concerns:

- Quantification of communication changes in Figure 3g: The claim that irradiation increases cellular communication is not visually clear. A quantifiable metric should be provided to support this conclusion.
- Use of scRNA-seq velocity: The study could benefit from using RNA velocity analysis to complement the pseudotime course and provide insights into lineage trajectories.
- Word cloud representation in Figure 3e: The use of a word cloud may not be the most effective way to represent the results. The authors should consider alternative visualizations.
- Axis labeling in Figure 3f: The axis measure is unclear; "relative contribution" is insufficient. The specific metric should be defined.
- Statistical significance in Figure 4L: There is no significant difference between the DMSO and JAG1 conditions (only between DBZ and JAG1). This should be discussed.
- Statistical significance in Figure 5G: A difference is mentioned in the text, but statistical significance between DMSO and JAG1 conditions should be explicitly stated.
- Inconsistency in figure labeling: There is a mix of capital and small letters in figure labels. Consistency should be maintained.

We thank the reviewers for their interest in our work and for the insightful comments, which have helped us enhance the significance and clarity of our results. Below, we provide our responses in blue, along with the corresponding text and figure references. Overall, we conducted new single-cell RNA-seq analyses, which are now included in the main Figures 1-3 and Appendix. Additionally, we performed further experiments, including western blots, to validate the upregulation of Notch signaling following irradiation and used publicly available online transcriptomic datasets to support our single-cell RNA-sequencing analysis.

Referee #1:

In this study, Cinat and colleagues utilize single-cell RNA sequencing to profile salivary gland organoids, identifying key cellular populations and signaling pathways. Their findings highlight Sox9- and Itgb1/Cd44-expressing cells as crucial stem/progenitor populations, with Notch signaling playing a central role in self-renewal and migration following irradiation. These results extend to other glandular tissues, emphasizing the conserved function of Notch signaling in stem/progenitor cell-mediated regeneration. Understanding the mechanisms underlying salivary gland regeneration is essential for improving the quality of life of cancer patients and survivors undergoing radiotherapy. While this study is both novel and timely, several conceptual and experimental aspects require further clarification.

Figure 1A: The authors mention that organoids are passaged; however, this is not depicted in the corresponding cartoon.

We apologize for the inaccuracy. We have now improved Figure 1A and included the passaging in the timeline.

Figure 1D-H: Specific clusters are identified using markers. However, employing a set of lineage-specific signatures combined with the AddModuleScore function from Seurat could provide a more unbiased approach. Additionally, to assess how closely salivary gland organoids resemble actual tissue, integrating different datasets could help determine how well cellular heterogeneity is recapitulated.

We thank the reviewer for the useful suggestion. While the recommended approach is effective for identifying known populations such as basal or luminal ducts, we believe that identifying rarer populations, such as stem/progenitor cells, is better achieved using the top significant genes identified with the FindAllMarkers function of Seurat. However, as suggested, to assess whether the organoid populations correspond to other cells in the tissue, we have now used gene signatures from the dataset GSE150327 (Hauser *et al*, 2020), in which the authors characterized mouse salivary glands at different ages: 1 day old (P1), 30 days old (P30), and adult. As shown in Appendix Figure S1A, our annotations strongly correlate with the respective cell populations identified in that study. We have updated the result section accordingly (page 3, lines 24-30):

“Additionally, using cell type-specific gene signatures extrapolated from the salivary gland tissue dataset GSE150327 (Hauser et al, 2020), we confirmed the identity of the luminal, basal, pro-acinar, and myoepithelial-like clusters, which closely aligned with their respective tissue-specific signature genes (Appendix Fig. S1A). Interestingly, stem/progenitor I and II populations showed strong correlation with intercalated ducts and Ascl3⁺ ducts, respectively

(Appendix Fig. S1A), regions previously proposed to harbor stem/progenitor populations (Rocchi et al, 2021; Rugel-Stahl et al, 2012)."

Moreover, as shown in Appendix Figure S2A and B, we have now integrated the organoid single-cell sequencing datasets with our salivary gland tissue dataset and the one from E-MTAB-13374 (McKendrick et al, 2023). Although the organoids clustered close to their respective tissue populations, we observed distinct differences between organoids and tissue. In particular, organoids showed an enrichment of early markers such as *Epcam*, *Cd24a*, *Krt14*, and *Krt6a*, confirming their more immature features. This supports the idea that organoid cultures are enriched for specific cell populations with a more stem cell-like phenotype, likely induced by the high concentration of embryonic laminin in the Matrigel used to culture the organoids (Hughes et al, 2010). However, organoids can form more differentiated structures when placed under differentiation conditions, as already demonstrated in previous studies (Nanduri et al, 2014; Yoon et al, 2022; Maimets et al, 2016). We have included this observation in the result section (page 4, lines 4-10):

"To further assess the similarity between mSGOs and adult salivary gland tissue, we integrated the mSGOs datasets with our salivary gland tissue dataset and the one from E-MTAB-13374 (McKendrick et al, 2023) (Appendix Fig. S2A). As expected, although organoid populations clustered closely with their respective tissue populations, mSGOs showed enrichment for progenitor-associated markers, such as Epcam, Cd24a, Krt6a and Krt14, and lacked more differentiated cell types, including acinar, macrophages and endothelial populations (Appendix Fig. SA, B)."

Overall, these data confirm an enrichment of salivary gland stem/progenitor cells within younger mSGOs and the presence of more differentiated cell types in older mSGOs. Nevertheless, mSGOs retain epithelial cells and a more progenitor-like phenotype compared to native tissue. Additionally, these findings reiterate the presence of potential salivary gland stem cell-like populations within the ductal compartment of the salivary glands."

Moreover, we have addressed in the discussion (page 9, lines 37-43) how the development of more complex organoid models, containing not only "immature" epithelial cells but also other cell types, could help further elucidate the function of these cells in a more physiological context:

"Moreover, since these organoids are primarily composed of more "immature" epithelial cells, likely due to the high concentration of embryonic laminin in the Matrigel used for their culture (Hughes et al, 2010), developing more physiologically relevant organoid models, such as air liquid interface (ALI) organoids that incorporate stromal and immune components (Santos et al, 2024), could provide deeper insights into the regulation of these cells and their response to irradiation within a more representative tissue context."

Figure 1F-G: In the scRNA-seq analysis, expression is relative to the other detected clusters. Showing a bubble plot from the integrated data would help support the claims about more mature and primitive states.

We thank the reviewer for the suggestion. We have included the requested dot plot in Appendix Figure S3D and used this integration for downstream RNA velocity and pseudotime analysis (page 4, lines 47-48 and page 5, lines 1-3). From the dot plot, it is interesting to observe that Day 11 organoids indeed express higher levels of more

differentiated markers, such as those found in pro-acinar cells, as well as luminal duct and basal cells, which show higher expression of *Krt7* and *Krt8*, markers typical of luminal ducts.

“To further support this notion, we integrated the 7-day and 11-day mSGO datasets (Fig. 2B; Appendix Fig. S3D) and performed unbiased RNA velocity and pseudotime analyses (Fig. 2D, E). Notably, cluster analysis and RNA velocity confirmed spontaneous differentiation over time, as evidenced by the directionality of the velocity flow (Fig. 2D) and the enrichment of more differentiated markers within the 11-day mSGO populations (Appendix Fig. S3D).”

Figure 1H: Is this panel showing a reclustering or simply a zoomed-in view? If the goal is to increase resolution, a reclustering approach would be more appropriate.

This figure panel indeed shows reclustering. We apologize if that was unclear, and we have now specified this in the legend of Figure 1H.

Figure 1N: The specificity of CD164 expression should be clarified, as it appears ubiquitously expressed in the provided image, which contrasts with the scRNA-seq data.

We appreciate the reviewer’s insightful observation. Indeed, CD164 appeared more ubiquitously expressed at the protein level than our single-cell RNA-seq data suggested. Nevertheless, we observed that SOX9⁺ cells showed higher levels of CD164 expression. As mentioned in the manuscript (page 4, lines 23-25), CD164 has recently been identified as a potential key marker for various human stem cells, including hematopoietic and skeletal stem cells (Watt *et al*, 2021). Although studies on CD164 expression in mice are very limited, we believe that with further validations, the co-expression of SOX9 and CD164 may serve as a potential strategy for identifying salivary gland stem/progenitor cells in future studies. However, we acknowledge the discrepancy between gene and protein expression and have included this observation in the results section (page 5, lines 15-21):

“Immunofluorescence staining showed a more ubiquitous expression of CD164 compared to scRNA-seq data; however, several SOX9-expressing cells exhibited notably higher CD164 levels than other cells (Fig. 1N). Overall, our data establish the presence of distinct cell populations within our mSGOs and provide new insights into salivary gland stem/progenitor cell populations and their markers.”

Figure 2A: The motifs appear specifically enriched at day 7 compared to day 11 in the ATAC-seq data. Including a bubble plot for these genes at day 11 would help contextualize these findings.

We thank the reviewer for the suggestion. We have now added an analysis of the ATAC-seq data from the merged Day 7 and Day 11 datasets to assess how the expression of transcription factor motifs changes over time. Moreover, we have included more stem cell-related transcription factor motifs, such as NF-Y and AP2. We have updated the result section accordingly (page 4, lines 42-47):

*“Specifically, motifs associated with KLF, TEAD, NF-Y, RUNX, AP2, SOX and FOX, which are known regulators of embryonic and stem cell-related processes (Fu *et al*, 2021; Sarkar & Hochedlinger, 2013; Jiang *et al*, 2008; Currey *et al*, 2021; Rigillo *et al*, 2021; Kim *et al*, 2014), were highly enriched in the stem/progenitor clusters, particularly in 7-day mSGOs (Fig. 2A), suggesting enhanced stem cell features and supporting the more immature profile of younger mSGOs.”*

Figure 2B: The presence of a day-dependent cluster suggests a potential batch effect. Improving data integration could address this issue.

We thank the reviewer for the remark. We agree that the presence of different clustering, even though the populations appear to be “the same,” might seem unexpected. To double-check this, we re-ran the data integration using several methods, including Seurat-CCA, Seurat-RPCA, and Harmony, with and without regressing out cell cycle effects. All methods led to similar results, particularly Seurat-CCA and Harmony. Ultimately, we chose the Harmony integration because Seurat-CCA integration method tended to “overcorrect”, resulting in the loss of some biological signal, whereas Harmony provided a more balanced integration for our dataset. In response to a comment from Reviewer 2, we also examined cell cycle distribution and observed differences between the two datasets. Therefore, we now present the analysis using Harmony integration with cell cycle regression in Figure 2C, which improved alignment of the populations while maintaining some separation between day 7 and day 11. The observed clustering differences does not seem primarily due to batch effects, but rather reflect genuine phenotypic shifts in the cells. As shown in previous figures, Day 11 clusters exhibit a clear increase in the expression of more differentiated markers, indicating that the cell types have undergone significant changes.

Figure 2C-D: The authors perform pseudotime analysis, which is conceptually complex given the two distinct time points. One would expect stem/progenitor cells to give rise to differentiated cells at each time point. Improved data integration could help resolve this issue. Another limitation is the method used to define the root of the pseudotime trajectory. Employing more unbiased approaches, such as RNA velocity and CellRank, could improve the identification of initial and terminal states.

We thank the reviewer for the insightful comment. Following also the suggestion of Reviewer 2, we have performed RNA velocity analysis using velocity and scVelo (as now described in the Methods section) and added the new results in Figure 2D. Additionally, we have conducted a new pseudotime analysis using Monocle3 (Figure 2E). An interesting observation is that one of the starting points originates not from the stem/progenitor II population, as we originally thought, but from basal cells. This finding aligns with previous studies (May *et al*, 2018; Kwak & Ghazizadeh, 2015; Kwak *et al*, 2018), which report that basal cells (*Krt14+*) exhibit progenitor-like features and have the ability to replace other types of cells after damage. Although we acknowledge that this observation requires further validation in future studies, it provides a potential explanation for the relatively high organoid-forming efficiency observed after passaging (higher than based solely on stem/progenitor cell populations) and opens interesting new avenues for research. We have updated the result section, including the new analysis (page 5, lines 1-11):

“To further support this notion, we integrated the 7-day and 11-day mSGO datasets (Fig. 2B; Appendix Fig. S3D) and performed unbiased RNA velocity and pseudotime analyses (Fig. 2D, E). Notably, cluster analysis and RNA velocity confirmed spontaneous differentiation over time, as evidenced by the directionality of the velocity flow (Fig. 2D) and the enrichment of more differentiated markers within the 11-day mSGO populations (Appendix Fig. S3D).

Based on RNA velocity analysis, we identified two precursor populations in the 7-day mSGOs: basal cells and stem/progenitor I cells (Fig. 2D). Pseudotime analysis further revealed that while stem/progenitor I cells appear to give rise to more duct-like cells, basal cells progressed towards the stem/progenitor cell II and pro-acinar populations, suggesting potential plasticity and de-differentiation capacity of these cells in vitro (Fig. 2E). This finding

aligns with previous studies (Kwak & Ghazizadeh, 2015; Kwak et al, 2018; May et al, 2018), which report that KRT14⁺ basal cells exhibit progenitor-like features and can replace other type of cells after damage.

And discussed this in the discussion section (page 10, lines 13-22):

“Interestingly, although Itgb1/Cd44-expressing cells exhibited clear stem cell properties, RNA velocity and pseudotime analyses identified basal cells as precursor cells for this population. This suggests a potential degree of cellular plasticity and regenerative capability within basal cells. This finding is consistent with previous studies (May et al, 2018; Kwak et al, 2018) demonstrating that salivary gland basal cells can exhibit progenitor-like behavior and contribute to tissue regeneration following damage. While further studies are needed to validate these observations, this opens the possibility that, similar to basal cells in the airway and mammary glands (Ma et al., 2024; Han et al., 2022), salivary gland basal cells may also retain stem/progenitor characteristics and latent plasticity that can be reactivated under specific conditions such as stress or (radiation) injury.”

Figure 2E: The bulk RNA-seq analysis of salivary gland organoids at days 5, 7, and 11 shows specific lineage and proliferation markers. The authors suggest an enrichment of stem/progenitor cells at day 7; however, the data indicate that days 7 and 11 are quite similar, whereas day 5 appears distinct. Statistical analyses should be applied to confirm whether differences between days 7 and 11 are significant.

We thank the reviewer for the comment. In light of the new analyses, including ATAC-seq for Day 7 and Day 11, annotation of the integrated dataset, and RNA velocity/pseudotime analysis, we found that this panel had become redundant and therefore decided to remove it. Specifically, as noted in previous responses, Figure 2A shows that ATAC-seq revealed an enrichment of stem cell-related transcriptomic factors at Day 7. Additionally, cluster annotation of the integrated dataset (Appendix Fig. S3D) demonstrated that 7-day mSGOs expressed more immature markers, while 11-day mSGOs showed a clear enrichment of differentiation markers (supporting the interpretation that integration differences reflect biological variation). Furthermore, RNA velocity and pseudotime analysis indicated a directional flow from Day 7 toward Day 11 populations (Figures 2D, 2E). Together, these observations confirm that younger organoids represent a more primitive cellular state. Additionally, we now performed deconvolution analysis using the bulk RNA-seq data of 5-day organoids (day of irradiation), which showed a high degree of similarity to 7-day organoids (Appendix Fig. S5A), as elaborated in a later response.

Figures 1F-I: The authors state that CD29 and CD24 are highly expressed in the stem/progenitor cluster. However, they are also prominently expressed in the basal and cycling clusters, making it challenging to attribute higher organoid formation efficiency solely to this cluster.

We agree with the reviewer that CD29 and CD24 are indeed expressed across most cell populations, as also shown in both our FACS data and previous studies (Nanduri *et al*, 2014). However, as shown in Appendix Figure 4A, the Stem/Progenitor I and Stem/Progenitor II populations display the highest expression levels of these markers. For this reason, we choose to select only the high-expressing populations from our FACS analysis. However, as also shown for CD29/CD44-expressing cells, we acknowledge that more subpopulations might exist and that future studies should aim at identifying more

specific markers to better isolate and characterize these stem/progenitor cells. We have addressed this point in the discussion section (page 11, lines 7-14):

“Furthermore, efforts should be made to identify more specific surface markers for these populations, as CD24 and CD29 expression was detected across a broad range of cells. Lastly, although both male and female patients were included in the study, the use of only female mice may limit the generalizability of the findings; therefore, incorporating male mice in future studies could help avoid potential sex-related biases. However, this consideration is mainly relevant for basic mouse studies, as male mice possess an additional glandular compartment, the glandular convoluted tubules, not present in human salivary glands of female mice.”

Figure 2M: The authors assess the migratory potential of 2D salivary gland organoids, but the analysis appears qualitative, as different populations are not compared. Would this differ in organoids at day 11? Additionally, CD44+ cells are reported at the migratory front, but are these cells also proliferative?

We thank the reviewer for the remark. To our knowledge, the scratch assay has not previously been performed on salivary gland cells (or any other normal epithelial tissue). With this panel, we aimed to validate the use of this assay in this context and demonstrate its feasibility. We have now also included the analysis for Day 11, which interestingly shows reduced cell migration compared to the earlier time point. However, also in this case, CD44+ cells accumulate at the migratory front, although to a lesser extent compared to Day 7, as shown in Appendix Figure S4H. We have updated the result section accordingly (page 5, lines 44-48 and page 6, lines 1-3):

“Notably, these salivary gland cultures, containing the primary cell populations found in the 3D organoid model (Appendix Fig. S4G), exhibited a clear migratory potential and the ability to fill the wound over time (Fig. 2M, N). However, salivary gland cells from day 11 exhibited a significantly reduced migration capacity compared to day 7 (Fig. 2N), likely due to their more differentiated phenotype, as evidenced by the loss of stem cell features and upregulation of pro-acinar markers (Fig. 2A; Appendix Fig. S3D).”

To assess proliferation, we performed KI67 staining following the scratch assay. However, as noted in the result section, the assay was conducted under starvation conditions specifically to minimize the influence of proliferation. We included KI67 staining of salivary gland and thyroid gland cells 18 hours after the scratch under starvation as a validation step, as now shown in Appendix Figure S4F (for salivary gland cells) and Appendix Figure S9D (for thyroid gland cells). Very few cells are proliferating under these conditions, both at the migratory front and within the cell monolayer, supporting the conclusion that CD44+ cells close the wound through migration rather than proliferation.

Figure 3A: After characterizing cell clusters in salivary gland organoids, the authors transition to modeling regeneration via photon and proton irradiation. The experimental design is somewhat unexpected, as irradiation is performed at day 5, a time point that was not previously characterized. Given the prior focus on day 7, it would be more logical to irradiate at day 7 and analyze responses at day 11. Performing scRNA-seq at day 5 could help resolve this discrepancy.

We thank the reviewer for this comment and observation. We chose to irradiate at this specific time because it has been well characterized in several previous studies (Peng *et al*,

2020; Cinat *et al*, 2024, 2023; Soto-Gamez *et al*, 2024). Moreover, as shown in earlier works (Cinat *et al*, 2023; Peng *et al*, 2020; Soto-Gamez *et al*, 2024), organoids progressively undergo senescence and cell cycle arrest over time (with a pick around day 12), processes that are known to significantly influence cellular phenotype and transcriptional profiles. Furthermore, in a study currently under revision (Cinat *et al*, 2024), we observed strong interferon responses at Day 11, along with substantial shifts in cell populations following irradiation, suggesting that irradiation at later time points introduces significant variability. For these reasons, we focused our analysis on earlier time points. Additionally, we do not expect major differences between Day 5 and Day 7, as organoid size and morphology remain largely unchanged between these two time points. To further confirm this, we performed cell deconvolution analysis of bulk RNA-seq data from Day 5 organoids (time of irradiation) and Day 7 organoids (as a control dataset) using our single-cell RNA-seq dataset from day 7 organoids. As shown in Appendix Figure S5A, CIBERSORTx outputs showed strong correlation with our single-cell RNA-seq data and indicated similar proportions of cell types at both time points, with Day 5 organoids exhibiting a slightly higher abundance of stem/progenitor and cycling cells.

Figure 3B: The authors claim that photon and proton irradiation do not induce significant differences and analyze the data collectively. However, the UMAP suggests otherwise. Would it be possible to show the proportions of cells in each cluster under different conditions (e.g., using scCODA)?

We thank the reviewer for the observation. What we meant here is that photon and proton irradiation did not lead to differences in the types of cells present. We apologize if that was unclear and have now rephrased this sentence (page 6, lines 18-20):

“Cluster annotation was executed by using the previously identified markers (Fig. 1F), and the presence of similar cell types compared to control samples was identified (Appendix Fig. S5C, D).”

Cell proportions are shown in Appendix Fig. S5D. While we observed some changes following irradiation, the differences in cell proportions between photon and proton irradiation were minimal. However, we agree with the reviewer that some differences at the gene expression level were present, as shown by the GSEA analysis (Appendix Fig. S5E) and some experiments later in the manuscript.

Figure 3D: The authors perform GSEA to identify enriched pathways in irradiated samples. While GSEA is valuable for generating hypotheses, experimental validation of some observed correlations would strengthen the conclusions.

Although we partially agree with the reviewer, the analysis presented in Figure 3D (now Appendix Fig. S5E) was performed as an unbiased approach to assess the upregulation of stem cell-related GO terms and Notch-related genes. Indeed, these genes were found to be upregulated across several biological processes following irradiation, a finding we further validate later in the manuscript. Additionally, pathways such as DNA methylation, mitochondrial dysfunction, and inflammation, which were also shown to be upregulated after irradiation, have been shown to play a role in stem cell function in our previous studies (Cinat *et al*, 2024, 2023).

Figure 3F: The authors present the relative contribution of ligand-receptor interactions in the control condition. Showing how these interactions change upon irradiation would enhance the panel's message.

Figures 3I-J: The authors conclude that Notch-related genes (Jag1, Jag2, Notch1) are upregulated upon irradiation. Are these differences statistically significant?

We thank the reviewer for the useful comments. To address both points in a single panel, we performed a new analysis using CellChat and replaced the previous panel. This analysis (shown in Figure 3I) shows how ligand-receptor interactions change between control and irradiation conditions. Additionally, it includes a differential expression analysis highlighting significantly upregulated ligand-receptor interactions following irradiation (including Notch genes in red). We added the interpretation of this new analysis in the result section (page 6, lines 45-48 and page 7 lines 1-4):

“Differential expression analysis of ligand-receptor interactions using CellChat confirmed the upregulation of several signaling pathways following irradiation, including Notch signaling, which was mediated by Notch1-Jag1 and Notch1-Jag2 interactions (Fig. 3I). Furthermore, compared to other ligand-receptor pairs, Notch-related genes appeared the more broadly expressed among the various cell populations (Fig. 3I; Appendix Fig. S6D), further supporting a key role for Notch signaling in maintaining cell function under regenerative conditions.”

Is there supporting protein-level evidence? Additionally, in the 3D organoid context, where are the sending and receiving cells located? Are there detectable differences in cleaved Notch protein levels or distribution?

To address the second part of this comment, we have now performed western blot analysis for activated Notch1 and Jag1 following irradiation (Figure 3J and Appendix Fig. S6F). Although we observed some variability across biological and experimental replicates, we consistently detected a significant upregulation of both proteins. We have now included these results in Figures 3J and Appendix Fig. S6F and commented on it in the result section (page 7, lines 4-6):

“Western blot analysis further confirmed these findings at the protein level, showing a significant upregulation of activated NOTCH1 and JAG1 2 days post-irradiation (Fig. 3J; Appendix Fig. S6F).”

In addition, we have conducted immunofluorescence staining for Notch1 and Jag1, which is provided below for review purposes and not included in the final manuscript. Notch1 was found to be ubiquitously expressed throughout the organoid, whereas Jag1 expression was predominantly localized at the periphery. While this pattern was generally consistent, some variation was observed between individual organoids.

7-day old mSGOs

Figure 4: To investigate Notch signaling in salivary gland organoid regeneration, the authors inhibit its activation using DBZ. However, they do not provide direct evidence that their treatment effectively inhibits the pathway. Additionally, while functional assays are performed, the transcriptional and cell identity changes induced by Notch inhibition remain unclear, complicating result interpretation.

We thank the reviewer for the comment. We validated the expression of Notch-related target genes following DBZ treatment in both salivary and thyroid gland cells, demonstrating a significant downregulation of Notch target genes, particularly *Hey1* and *Hes1*, as shown in Appendix Figure 7A (salivary gland) and Appendix Figure 9A (thyroid gland). This confirms the effectiveness of the dose used, which is also in line with previous data and other organoid studies where they all used DBZ 1 μ M to inhibit the pathway (Kessler *et al*, 2015; Xie *et al*, 2018; Bustamante-Madrid *et al*, 2024).

Additionally, we have now assessed the expression of differentiation markers by rt-qPCR on salivary gland organoids treated with DBZ (Appendix Figure 7B). Interestingly, while the expression of duct cell markers decreased, acinar cell markers such as *Prol1* and *Aqp5* were upregulated following DBZ treatment. This confirms that DBZ treatment induces premature differentiation, specifically towards an acinar phenotype. This is also in line with the more lobular phenotype of differentiated organoids shown in Figure 4E. We added this observation in the result section (page 7, lines 24-28):

“Notably, Notch inhibition (Appendix Fig. S7A) impaired mSGO growth (Fig. 4B) and significantly decreased the self-renewal capacity of salivary gland stem/progenitor cells, shown as a reduction of OFE and population doubling (PD) (Fig. 4C). Moreover, DBZ-treated mSGOs showed increased expression of acinar markers Prol1 and Aqp5, suggesting premature differentiation toward an acinar-like phenotype (Appendix Fig. S7B).”

Figure 4E: The organoids treated with and without DBZ appear different from day 0, which is unexpected. Including whole-well images and quantifications of circularity or budding structures would provide a clearer picture.

Since all Day 0 (start of differentiation) organoids appeared morphologically similar, we selected representative images from the various conditions. We apologize if the images used previously did not appear consistent, potentially leading to confusion. To improve clarity and provide a more accurate representation, we have now selected a more representative,

rounded-shaped organoid for the DBZ condition and included whole-well images in Appendix Figure 7C, along with morphological quantification of lobular vs round organoids as requested.

Figure 4L: There appears to be a discrepancy between Figures 4L and 4C. Can the authors clarify this? Additionally, in the DMSO-treated condition, the control values are consistently set to 1, suggesting no variability. How were statistical analyses applied in this context? This data processing approach risks inflating small fold-change differences into statistically significant results.

The discrepancy the reviewer refers to may have arisen from differences in the control conditions due to the distinct experimental setups, as outlined in Figures 4A and 4J. In the first experiment, DBZ was added from Day 0 (single cell state) to assess its impact on organoid growth and self-renewal capacity. In contrast, in the experiment shown in Figure 4J, organoids were treated with DBZ after irradiation, when organoids were fully formed, to assess the involvement of Notch signaling in self-renewal post-irradiation. In this context, while there might be slight changes in organoid size, since the organoids are already fully formed, the control organoids appeared more similar across conditions. We have clarified this in the result section (page 8, lines 2-8):

“To address if surviving stem/progenitor cells rely on Notch to maintain their self-renewal capacity after irradiation, mSGOs were irradiated at day 5 and treated with either the Notch inhibitor DBZ or the Notch activator JAG1 from day 7. Treated 11-day mSGOs were collected, and self-renewal capacity was assessed (Fig. 4J). It is important to note that in this assay, organoids were treated when fully formed, rather than as single cells as done in non-irradiated conditions (Fig. 4A).”

We agree with the reviewer that using FC data may influence data interpretation. In line with the reviewer’s suggestion, we have now updated the graphs in Figure 4L and Appendix Figure S7H to display the raw data. Each biological replicate is now indicated using distinct dot shapes to improve clarity. For statistical analysis, we have performed a one-way repeated-measures ANOVA followed by Tukey’s post hoc test, which confirmed significant differences between DMSO-DBZ and DBZ-JAG1 treatments in irradiated samples. However, due to high variability, the comparison between DMSO and JAG1 did not reach statistical significance. Accordingly, we have updated the results section (page 8, lines 12–15), indicating a trend toward increased OFE rather than a statistically significant change.

“Notably, while JAG1 treatment showed a trend toward increased OFE in photon-irradiated samples (Figure 4K, L), no changes were detected in proton-irradiated mSGOs (Appendix Fig. S7H), which already exhibited higher levels of Notch signaling than photon at the single-cell level (Appendix Fig. S6E).”

Figures 5K, N, P: Similar to the previous comment, it is unexpected that all controls are normalized to a fold-change of 1. How were statistical analyses performed under this assumption? Showing raw data instead of fold changes might be more informative and would allow conclusions to be adapted accordingly.

Similar to the previous comment, FC was used initially due to the high variability observed across different patient samples. However, we have now included the raw data and performed a paired t-test to provide a more accurate statistical analysis.

Referee #2:

This study provides valuable insights into the cellular composition and regenerative potential of mouse submandibular salivary gland organoids (mSGOs) through single-cell RNA sequencing (scRNA-seq) and functional assays. The analysis identifies distinct cell populations in mSGOs at 7 and 11 days of culture, including two stem/progenitor cell populations, basal and luminal duct cells, and differentiated cell types. The 7-day organoids exhibit a more stem/progenitor-like phenotype, while the 11-day organoids show increased differentiation. Functional assays confirm that stem/progenitor cells from 7-day mSGOs have self-renewal capabilities. Additionally, irradiation studies show that stem/progenitor cells surviving radiation exhibit increased regenerative activity and Notch signaling, highlighting their role in tissue repair. The study addresses important questions regarding the plasticity and resilience of salivary gland cells. However, limitations such as the lack of in vivo and in vitro integration, the absence of cell cycle normalization, and potential sex bias should be considered.

Major Concerns:

- Integration of in vivo and in vitro datasets: To further strengthen the comparison between in vitro and in vivo conditions, it would be valuable to integrate the datasets and perform a direct comparative analysis between the mSGO-derived populations and their corresponding counterparts in native salivary gland tissue. While the study highlights similarities between mSGOs and in vivo tissue, a systematic assessment of transcriptional differences between these conditions could provide deeper insights into how well the organoid model recapitulates the native environment. For instance, identifying differentially expressed genes between mSGO-derived stem/progenitor cells and their in vivo counterparts could reveal culture-induced adaptations or limitations of the in vitro system.

We thank the reviewer for this comment and fully agree that integrating our dataset with salivary gland tissue data would enhance the characterization of our cell populations. In line with this, and following also the suggestion of Reviewer 1, we have now integrated our salivary gland organoid datasets with a publicly available salivary gland tissue dataset from E-MTAB-13374 (McKendrick et al., 2023) and ours. As shown in Appendix Figure S2A and B, organoids exhibit enrichment for early epithelial markers such as *Epcam*, *Cd24a*, *Krt14*, and *Krt6a*, consistent with a more immature/stem-like phenotype. These findings support the idea that organoid cultures are enriched for specific progenitor populations, a phenotype that is progressively lost upon differentiation, consistent with previous studies (Nanduri et al., 2014; Yoon et al., 2022; Maimets et al., 2016). We have included this observation in the result section (page 4, lines 4-10):

“To further assess the similarity between mSGOs and adult salivary gland tissue, we integrated the mSGOs datasets with our salivary gland tissue dataset and the one from E-MTAB-13374 (McKendrick et al, 2023) (Appendix Fig. S2A). As expected, although organoid populations clustered closely with their respective tissue populations, mSGOs showed enrichment for progenitor-associated markers, such as Epcam, Cd24a, Krt6a, and Krt14, and lacked more differentiated cell types, including acinar, macrophages, and endothelial populations (Appendix Fig. SA, B).”

Overall, these data confirm an enrichment of salivary gland stem/progenitor cells within younger mSGOs and the presence of more differentiated cell types in older mSGOs. Nevertheless, mSGOs retain epithelial cells and a more progenitor-like phenotype compared to native tissue. Additionally, these findings reiterate the presence of potential salivary gland stem cell-like populations within the ductal compartment of the salivary glands.”

- Cell cycle normalization: The presence of a cycling population suggests that normalization should be performed to ensure that clustering and gene expression differences are not primarily driven by proliferative status.

We thank the reviewer for the observation. Although a distinct and very small cell cycle-related cluster was detected, the remaining clusters showed similar (and very low) expression of cell cycle-related genes (day 7 shown below for review purposes). Therefore, we chose to keep the original clustering.

However, when integrating the Day 7 and Day 11 datasets, we did observe differences in cell cycle distribution between timepoints. To account for this, we regressed out cell cycle effects in the integrated dataset prior to performing RNA velocity and pseudotime analyses. This is now described in the method section and included in the data presented in Figure 2.

- Sex bias in study design: Only female mice were used. The authors should provide a rationale for this choice and discuss potential limitations in terms of generalizability.

We agree with the reviewer that the use of only female mice represents a limitation and should be acknowledged. The rationale behind this choice is that male mice possess an extra sex-specific gland within the submandibular gland (not present in humans), the granular convoluted tubules, which produce growth factors and hormones that significantly alter the cell culture environment. These factors have been shown to interfere with organoid formation, limiting the ability of cells from male mice to reliably generate organoids. However, it is important to note that for the human organoid experiments, samples were derived from both male and female patients. We added the reason behind this choice in the methods (page 11, lines 33-34):

“Mice. Given the known anatomical differences between male and female mice, particularly the presence of granular convoluted tubules within the submandibular glands of males, which are absent in humans, only female mice were used for this study.”

And commented about it in the discussion (page 11, lines 18-23):

“Lastly, although both male and female patients were included in the study, the use of only female mice may limit the generalizability of the findings; therefore, incorporating male mice in future studies could help avoid potential sex-related biases. However, this consideration is mainly relevant for basic mouse studies, as male mice possess an additional glandular compartment, the glandular convoluted tubules, not present in human salivary glands or in female mice.”

Minor Concerns:

- Quantification of communication changes in Figure 3g: The claim that irradiation increases cellular communication is not visually clear. A quantifiable metric should be provided to support this conclusion.

We agree with the reviewer that estimating the number of interactions from the Chord Plots is difficult. To address this, we have now included a quantification of the number of interactions in Figure 3G, as well as an analysis of interaction strength in Appendix Figure S6B showing increased number of interactions and interaction strength after irradiation.

- Use of scRNA-seq velocity: The study could benefit from using RNA velocity analysis to complement the pseudotime course and provide insights into lineage trajectories.

We thank the reviewer for the useful insight. In response to this comment, and in line with the suggestions from Reviewer 1, we have now performed RNA velocity analysis using scVelo and pseudotime analysis using Monocle3. These new results have been incorporated into Figure 2 and in the result section (page 4, lines 47-48 and page 5, lines 1-13):

“To further support this notion, we integrated the 7-day and 11-day mSGO datasets (Fig. 2B; Appendix Fig. S3D) and performed unbiased RNA velocity and pseudotime analyses (Fig. 2D, E). Interestingly, RNA velocity revealed two precursor populations in the 7-day mSGOs: basal cells and stem/progenitor I cells (Fig. 2D). Based on these starting points, pseudotime analysis showed that while stem/progenitor I cells appear to give rise to more duct-like cells, basal cells progressed towards the stem/progenitor cell II and pro-acinar populations, suggesting plasticity and de-differentiation capacity of these cells in vitro (Fig. 2E). This finding aligns with previous studies (May et al, 2018), which report that KRT14+ basal cells exhibit progenitor-like features and can replace other type of cells after damage. Together, these data indicate an enrichment of stem/progenitor cells in 7-day mSGOs and highlight Sox9-, Krt14- and Itgb1-expressing cells as the most primitive cell types within our mSGOs.”

- Word cloud representation in Figure 3e: The use of a word cloud may not be the most effective way to represent the results. The authors should consider alternative visualizations.

We thank the reviewer for the suggestion. However, we believe that the word cloud effectively highlights the most enriched genes in a visually intuitive manner, offering a clearer visualization over traditional bar plots. For this reason, we have decided to keep this panel.

- Axis labeling in Figure 3f: The axis measure is unclear; "relative contribution" is insufficient. The specific metric should be defined.

We apologize if the axis label was unclear. This format is the default output generated by CellChat for this type of graph. However, we checked CellChat information and confirmed that the graph represents the proportion of each ligand-receptor (L–R) pair. We have now

clarified this in the axis label and specified the axis range based on the extracted graph data (now Figure 3E).

- Statistical significance in Figure 4L: There is no significant difference between the DMSO and JAG1 conditions (only between DBZ and JAG1). This should be discussed.

We agree with the reviewer. We have updated the result section, emphasizing that this is indeed not a significant change but rather a trend toward an increase (page 8, lines 39-42):

“Notably, while JAG1 treatment showed a trend toward increased OFE in photon-irradiated samples (Figure 4K, L), no changes were detected in proton-irradiated mSGOs (Appendix Fig. S7H), which already exhibited higher levels of Notch signaling at the single-cell level (Appendix Fig. S6E).”

- Statistical significance in Figure 5G: A difference is mentioned in the text, but statistical significance between DMSO and JAG1 conditions should be explicitly stated.

We apologize for the oversight. We have now re-analyzed the scratch assay using a larger area and included statistical significance where appropriate in Figure 5G.

- Inconsistency in figure labeling: There is a mix of capital and small letters in figure labels. Consistency should be maintained.

We apologize for the inconsistency. Figure labels have now been updated to align with the journal's formatting guidelines.

References

- Bustamante-Madrid P, Barbáchano A, Albandea-Rodríguez D, Rodríguez-Cobos J, Rodríguez-Salas N, Prieto I, Burgos A, Martínez de Villarreal J, Real FX, González-Sancho JM, *et al* (2024) Vitamin D opposes multilineage cell differentiation induced by Notch inhibition and BMP4 pathway activation in human colon organoids. *Cell Death Dis* 15: 1–16
- Cinat D, Souza AL De, Soto-gamez A, Jellema-de AL, Coppes RP & Barazzuol L (2023) Mitophagy induction improves salivary gland stem/progenitor cell function by reducing senescence after irradiation. *Radiother Oncol*: 110028
- Cinat D, van der Wal R, Baanstra M, Soto-gamez A, Jellema-De Bruin AL, Van Goethem MJ, van Vugt MATM, Barazzuol L & Coppes RP (2024) Derepression of transposable elements enhances interferon beta signaling and stem/progenitor cell activity after proton irradiation. *bioRxiv*
- Hauser BR, Aure MH, Kelly MC, Hoffman MP & Chibly AM (2020) Generation of a Single-Cell RNAseq Atlas of Murine Salivary Gland Development. *iScience* 23: 101838
- Hughes CS, Postovit LM & Lajoie GA (2010) Matrigel: a complex protein mixture required for optimal growth of cell culture. *Proteomics* 10: 1886–1890
- Kessler M, Hoffmann K, Brinkmann V, Thieck O, Jackisch S, Toelle B, Berger H, Mollenkopf HJ, Mangler M, Sehoul J, *et al* (2015) The Notch and Wnt pathways regulate stemness and differentiation in human fallopian tube organoids. *Nat Commun* 6

- Kwak M & Ghazizadeh S (2015) Analysis of histone H2BGFP retention in mouse submandibular gland reveals actively dividing stem cell populations. *Stem Cells Dev* 24: 565–574
- Kwak M, Ninche N, Klein S, Saur D & Ghazizadeh S (2018) c-Kit⁺ Cells in Adult Salivary Glands do not Function as Tissue Stem Cells. *Sci Rep* 8: 1–11
- Maimets M, Rocchi C, Bron R, Pringle S, Kuipers J, Giepmans BNG, Vries RGJ, Clevers H, de Haan G, van Os R, *et al* (2016) Long-Term In Vitro Expansion of Salivary Gland Stem Cells Driven by Wnt Signals. *Stem cell reports* 6: 150–162
- May AJ, Cruz-Pacheco N, Emmerson E, Gaylord EA, Seidel K, Nathan S, Muench MO, Klein OD & Knox SM (2018) Diverse progenitor cells preserve salivary gland ductal architecture after radiation-induced damage. *Dev* 145
- McKendrick JG, Jones GR, Elder SS, Watson E, T'Jonck W, Mercer E, Magalhaes MS, Rocchi C, Hegarty LM, Johnson AL, *et al* (2023) CSF1R-dependent macrophages in the salivary gland are essential for epithelial regeneration after radiation-induced injury. *Sci Immunol* 8
- Nanduri LSY, Baanstra M, Faber H, Rocchi C, Zwart E, De Haan G, Van Os R & Coppes RP (2014) Purification and Ex vivo expansion of fully functional salivary gland stem cells. *Stem Cell Reports* 3: 957–964
- Peng X, Wu Y, Brouwer U, van Vliet T, Wang B, Demaria M, Barazzuol L & Coppes RP (2020) Cellular senescence contributes to radiation-induced hyposalivation by affecting the stem/progenitor cell niche. *Cell Death Dis* 11: 1–11
- Rocchi C, Barazzuol L & Coppes RP (2021) The evolving definition of salivary gland stem cells. *npj Regen Med* 6: 1–8
- Rugel-Stahl A, Elliott ME & Ovitt CE (2012) Ascl3 marks adult progenitor cells of the mouse salivary gland. *Stem Cell Res* 8: 379–387
- Santos AJM, van Unen V, Lin Z, Chirieleison SM, Ha N, Batish A, Chan JE, Cedano J, Zhang ET, Mu Q, *et al* (2024) A human autoimmune organoid model reveals IL-7 function in coeliac disease. *Nature* 632: 401–410
- Soto-Gamez A, van Es M, Hageman E, Serna-Salas SA, Moshage H, Demaria M, Pringle S & Coppes RP (2024) Mesenchymal stem cell-derived HGF attenuates radiation-induced senescence in salivary glands via compensatory proliferation. *Radiother Oncol* 190: 109984
- Watt SM, Bühring HJ, Simmons PJ & Zannettino AWC (2021) The stem cell revolution: on the role of CD164 as a human stem cell marker. *npj Regen Med* 6: 1–6
- Xie Y, Park ES, Xiang D & Li Z (2018) Long-term organoid culture reveals enrichment of organoid-forming epithelial cells in the fimbrial portion of mouse fallopian tube. *Stem Cell Res* 32: 51–60
- Yoon YJ, Kim D, Tak KY, Hwang S, Kim J, Sim NS, Cho JM, Choi D, Ji Y, Hur JK, *et al* (2022) Salivary gland organoid culture maintains distinct glandular properties of murine and human major salivary glands. *Nat Commun* 13: 1–16

Dear Dr Coppes,

Thank you for submitting your revised manuscript (EMBOJ-2025-120401R) to The EMBO Journal, as well for your patience with our feedback. Your amended study was sent back to the referees for their scientific reassessment, and we have received the report by one of them, which I enclose below. Please note that while referee #2 was unfortunately not able to reevaluate your amended study at this time, we have editorially assessed your response to the critique by this expert and found the raised issues to be addressed satisfactorily. As you will see, the other reviewer states that the work has been substantially enhanced by the revisions and s/he is now in favour of publication, pending minor amendments.

Thus, we are pleased to inform you that your manuscript has been accepted in principle for publication in The EMBO Journal.

Please carefully consider the remaining minor point raised by referee #1 by adjusting the discussion and data presentation / statistical annotation where appropriate.

Also, we now need you to take care of a number of issues related to formatting and data presentation as detailed below, which should be addressed at re-submission.

Please contact me at any time if you have additional questions related to below points.

As you might have seen on our web page, every paper at the EMBO Journal now includes a 'Synopsis', displayed on the html and freely accessible to all readers. The synopsis includes a 'model' figure as well as 2-5 one-short-sentence bullet points that summarize the article. I would appreciate if you could provide this figure and the bullet points.

Thank you for giving us the chance to consider your manuscript for The EMBO Journal. I look forward to your final revision.

Again, please contact me at any time if you need any help or have further questions.

Best regards,

Daniel Klimmeck

>> Please add up to five keywords to your study.

>> Author Contributions: Remove the author contributions information from the manuscript text. Note that CRediT has replaced the traditional author contributions section as of now because it offers a systematic machine-readable author contributions format that allows for more effective research assessment. and use the free text boxes beneath each contributing author's name to add specific details on the author's contribution.

More information is available in our guide to authors.
<https://www.embopress.org/page/journal/14602075/authorguide>

>> Adjust the title of the 'Declaration of Interests' section to 'Disclosure and Competing Interests Statement'.

>> Rename the 'Material and methods' to 'Methods'.

>> Remove the figures from the main text file.

>> Correct the order of the manuscript sections as follows: Abstract / Keywords / Introduction / Results / Discussion / Methods / Data Availability / Acknowledgements / Disclosure and Competing Interests Statement / References / Main Figure Legends / Tables / Expanded View Figure Legends.

>> Figure callouts: Please ensure that the Fig 2C is called out in sequential order in the main text.

>> Appendix File with ToC: please add page numbers to the table of contents; please remove the reagents and tools table and upload it as a separate file using our template; renumber the table with the primers in the appendix Appendix Table S1 accordingly.

>> Reagents and Tools table: Please upload as a separate file using the existing template in the Guide For Authors, listing key reagents, experimental models, software and relevant equipment.

>> Please indicate redisplay of data from Fig 2M in the figure legend of Figure 4H. Recheck indication of redisplay of Western Blots within Figure 3J in the figure legend.

>> Recheck references for the bioRxiv entry Citat et al. (2024) and update the citation if in the meantime published as regular article.

>> BioRender: move the sentence about the use of BioRender from the Acknowledgments to a dedicated "Graphics" section in the Methods using this format:

Graphics:

(some of the... OR Figure #... OR synopsis) Graphics were created with BioRender.com.

>> Data availability section: please remove the referee token for the GEO datasets and make sure that the data are made publicly accessible. Add hyperlinks to the datasets.

>> Dataset EV legends: rename Tables EV1 and EV2 to make them Dataset EV1 and EV2; please remove the legends from the manuscript text and them to each dataset file.

>> Consider additional changes and comments from our production team as indicated below:

- Figure legends:

1. Please note that the figure 5C is mislabeled as figure 5V in the manuscript. This needs to be rectified.
2. Please note that information related to n is missing in the legends of figures 1I, 2N, 5C
3. Please note that the error bars are not defined in the legends of figures 2N, 5C

Please use the link below to submit your revision:

Referee #1:

In the revised version of the manuscript, Cinat and colleagues have addressed the majority of my previous comments and concerns. I appreciate the substantial effort invested in the revision, and I believe the manuscript has been significantly improved as a result. The additional analyses, particularly the integration of scRNA-seq datasets and the inclusion of RNA velocity and pseudotime analyses, have strengthened the conclusions. However, a few minor points remain:

-The role of Notch signaling in regulating stemness within the organoid system is now well supported. However, it remains unclear whether this activation is a specific response to injury or a more general feature of stem/progenitor maintenance. While transcriptional data indicates upregulation of Notch pathway components, the protein levels of JAG1 and activated NOTCH1 appear only modestly increased post-irradiation. This raises the possibility of additional regulatory layers, such as post-translational modifications or receptor-ligand dynamics, that may not be captured by bulk protein quantification. A brief discussion of this potential complexity would be valuable. Additionally, the inclusion of feature plots or violin plots for canonical Notch target genes (e.g., HES1, HEY1) in Figure S6D-E would further support the transcriptional activation of the pathway.

-As raised in the first round, the use of fold-change normalization for statistical testing can inflate significance and potentially mislead interpretation. This concern still applies to several figures in the revised manuscript (e.g., Fig. 2L, Fig. S7A-B, Fig. S8A). It is essential to clarify whether statistical tests were performed on raw, unnormalized data or on fold-change values. If the latter, the authors should consider reanalyzing these datasets using raw values. This clarification is important to ensure the robustness of the reported significance levels.

The authors addressed the remaining editorial issues.

Dear Dr Coppes,

Thank you for submitting the revised version of your manuscript. I have now evaluated your amended study and concluded that the remaining minor concerns have been sufficiently addressed.

I am thus pleased to inform you that your manuscript has been accepted for publication in the EMBO Journal.

On a different note, I would like to alert you that EMBO Press offers a format for a video-synopsis of work published with us, which essentially is a short, author-generated film explaining the core findings in hand drawings, and, as we believe, can be very useful to increase visibility of the work. Please see the following link for representative examples:
https://www.embopress.org/video_synopses
<https://www.embopress.org/doi/full/10.1038/s44318-025-00417-0>

Finally, we have noted that the submitted version of your article is also posted on the preprint platform bioRxiv. We would appreciate if you could alert bioRxiv on the acceptance of this manuscript at The EMBO Journal in order to allow for an update of the entry status. Thank you in advance!

Best regards,

Daniel Klimmeck

Daniel Klimmeck, PhD
Senior Editor
The EMBO Journal
EMBO
Postfach 1022-40

Meyerhofstrasse 1
D-69117 Heidelberg
contact@embojournal.org
